# Molecular insights into ago-allosteric modulation at cysteinyl leukotriene receptor 2

Mu Li[1,5], Xiaoling Bao[2,5], Wanbiao Chen[3,5], Yusheng Guo [1,5], Xiaomin Mao[1,5], Miaofang Xiao[1,5], Siqi Liu[1], Jiawei Li[3,4], Limin Zhao[4], Tiancai Chang[1], Fumei Zhong[1], Chongyuan Wang [3] ✉ & Heng Liu [1] ✉

Cysteinyl leukotriene receptor CysLT2R, which is activated by the endogenous cysteinyl leukotrienes (CysLTs) LTC4, LTD4, and LTE4, has emerged as a potential therapeutic target due to the involvement in various inflammatory diseases. Accumulating evidence indicates that CysLT2R is also involved in the pathogenesis of cardiovascular diseases and contribute to tumor progression in cancer. However, the structural basis underlying the ligand recognition and the receptor activation remains to be elucidated. Here, we present two cryo-electron microscopy (cryo-EM) structures of the human CysLT2R-$G_q$ complexes bound to LTC4 and LTD4. CysLTs are characterized as ago-allosteric modulators (ago-PAMs) of CysLT2R. Our structures reveal that CysLTs are recognized by a lipid-facing pocket above intracellular loop 2 (ICL2) near the cytoplasmic side of the receptor. Furthermore, a noncanonical activation mechanism exists between the allosteric binding pocket and the $G_q$-binding site. Our findings provide comprehensive insights into the recognition of CysLTs and $G_q$ protein signaling transduction by CysLT2R, which may facilitate rational design of drugs.

The proinflammatory lipid mediators CysLTs, including LTC4, LTD4, and LTE4, exert their effects through interactions with two G protein-coupled receptors (GPCRs), CysLT1R and CysLT2R[1–4]. CysLT1R plays an important role in the allergic and inflammatory disorders, such as asthma, rhinitis, urticaria and atopic dermatitis, and also functions in cardiovascular diseases and cancers[5–12]. Meanwhile, CysLT2R has been implicated in treatments of severe asthma, cardiovascular and neurodegenerative disorders, as well as several types of cancers[11–19]. Notably, CysLT2R has recently been identified as the endogenous receptor for ceramides, with report of ceramide-bound activated structure emphasizing its critical role in atherosclerosis[18]. Moreover, the distinct tissue distribution and expression profiles of CysLTRs suggest the different spectra of receptor-associated disorder and corresponding therapy strategies[2–4,13,20–22]. Extensive efforts have been made to discover drugs targeting CysLTRs. Among them, three CysLT1R-selective antagonists (montelukast, zafirlukast and pranlukast) have been approved to treat asthma with high potency[23]. However, side effects and nonrespondency have been reported[24]. Although CysLT2R-selective or dual antagonists have been introduced to treat severe asthma and brain injuries[25,26], the development of CysLT2R-targeted

[1]The Affiliated Traditional Chinese Medicine Hospital, GMU-GIBH Joint School of Life Sciences, The Guangdong-Hong Kong-Macao Joint Laboratory for Cell Fate Regulation and Diseases, Guangzhou Medical University, Guangzhou Municipal and Guangdong Provincial Key Laboratory of Protein Modification and Disease, State Key Laboratory of Respiratory Disease, Guangzhou, Guangdong, China. [2]Scientific Research Center of Guangzhou Medical University, Guangzhou, Guangdong, China. [3]Center for Human Tissues and Organs Degeneration, Faculty of Pharmaceutical Sciences, Shenzhen Institutes of Advanced Technology, Chinese Academy of Sciences, Shenzhen, Guangdong, China. [4]Department of Geriatric Medicine, Shenzhen Longhua District Central Hospital, Shenzhen, Guangdong, China. [5]These authors contributed equally: Mu Li, Xiaoling Bao, Wanbiao Chen, Yusheng Guo, Xiaomin Mao, Miaofang Xiao. ✉e-mail: cy.wang@siat.ac.cn; 2022991053@gzhmu.edu.cn

drugs still faces challenges due to the lack of understanding of ligand recognition and receptor activation mechanism triggered by CysLTs. To facilitate the development of efficient therapies for diseases related to CysLT2R, a range of crystal structures bound to antagonists have been determined, providing atomic-level structural insights into antagonist recognition and subtype selectivity[27]. However, the absence of CysLTs-bound CysLT2R structures has limited a complete mechanistic understanding of the endogenous agonist recognition and receptor activation.

In this study, we report two active structures of the human CysLT2R-$G_q$ complexes bound to LTC4 and LTD4. Intriguingly, our structures demonstrate that CysLTs interact with CysLT2R at allosteric binding pocket formed by TM3, TM4, TM5 and ICL2, which is distinct from the extracellular pocket occupied by ceramides and antagonists. While this manuscript was in preparation, a cryo-EM structure of LTD4-bound CysLT2R-$G_q$ complex was reported[28]. In that structure, LTD4 was modeled in the orthosteric binding pocket, similar to CysLT2R antagonists[27]. Therefore, to elucidate the orthosteric and allosteric action of chemically diverse agonists, here we also provide a comprehensive functional assay triggered by LTC4 and ceramide coordinately. CysLTs are characterized as ago-PAMs for CysLT2R. Together with mutational studies, the structural information provides a framework for understanding the ago-allosterism, noncanonical activation mechanism and signaling transduction of CysLT2R and a structural basis for designing new anti-CysLTs drugs.

## Results

### Structure determination and overall structures of CysLTs-CysLT2R-$G_q$ complexes

The CysLTs LTC4, LTD4, and LTE4 are a lipid family conjugated with a peptide moiety (Fig. 1a), which are derived from arachidonic acid[1,29]. Both CysLTRs recognize three CysLTs with different potencies, predominately activating $G_{q/11}$ (Fig. 1b), and also signaling through $G_{i/o}$[2].

To elucidate the binding mode and signaling profile of CysLTs to CysLT2R, we employed single-particle cryo-EM to determine two structures of CysLT2R-$G_q$ protein complexes bound to LTC4 and LTD4, respectively. In our structural studies, we used the wild-type human CysLT2R. We also used an engineered miniG$\alpha_q$ subunit[30] with the N-terminal 35 amino acids replaced by their corresponding residues in G$\alpha_i$, which has been successfully used to obtain cryo-EM structures of several other $G_q$-coupled GPCRs[31-33]. The engineered mini-G$\alpha_q$ construct exhibits functional activity comparable to that of wild-type G$\alpha_q$[31]. Owing to its enhanced tendency for forming stable complexes with GPCRs, the mini-G$\alpha_q$ variant, whose C-terminal α5 helix remains unchanged, effectively mimics wild-type G$\alpha_q$ in receptor coupling. Consequently, mini-G proteins have been serving as ideal tools for biophysical investigations of GPCRs in their active state[34]. Unless otherwise stated, $G_q$ refers to the engineered mini-G$\alpha_q$-βγ heterotrimer used for structural determination. To improve the stability of CysLT2R-$G_q$ complexes for cryo-EM studies, we assembled the complexes using the NanoBiT tethering strategy[35] together with a single-chain variable fragment (scFv16), which has been developed previously to stabilize the $G_i$ heterotrimer[36]. We prepared the CysLT2R-$G_q$ complexes in lauryl maltose neopentyl glycol (LMNG)/cholesterol hemisuccinate (CHS) detergent condition for cryo-EM studies (Supplementary Fig. 1a, b). Ultimately, LTC4- and LTD4-bound CysLT2R-$G_q$ complexes were determined to overall resolutions of 3.3 and 3.5 Å, respectively (Fig. 1c, d, Supplementary Fig. 2, Table 1). The high-quality cryo-EM maps enabled us to model the majority of CysLT2R, $G_q$ and scFV16 antibody in the structures. Apart from the N terminus and C terminus, a large part (232-237) of ICL3 was also not modeled due to weak density, indicating that ICL3 was not involved in the interaction with $G_q$ protein. However, ICL2, which is highly flexible in inactive structures, became visible upon ligand binding due to its direct interaction with $G_q$. Cholesterol molecules around CysLT2R

surface were modeled (Fig. 1c, d). Both structures are in active state with large outward movement of transmembrane helix 6 (TM6), which is a hallmark of class A GPCRs activation[37].

Surprisingly, no density corresponding to CysLTs were found in the orthosteric binding pocket of CysLT2R, contrary to previous predictions[27,38]. Instead, a CHS molecule was found in the cleft between TM4 and TM5 (Figs. 1c, d and 2a), which was occupied by antagonists in the previously reported inactive CysLT2R structures (Supplementary Fig. 3b)[27,38]. Based on our structural and functional studies, we concluded that the extracellular pocket of CysLT2R is not the native binding site for CysLTs, which explains the poor density of extracellular part of receptors (Supplementary Fig. 2g, o). Given the lipidic nature of CysLTs, we therefore hypothesized that they might bind to the membrane side of the 7TM bundle. Indeed, the featured CysLTs densities were observed at a flat pocket formed by TM3, TM4, TM5 and ICL2 with a distinct recognition mechanism in CysLT2R (Fig. 2a). Interestingly, two cholesterol molecules were modeled at the corresponding site in the ceramide-CysLT2R-$G_q$ structure (Fig. 2a), which supports that this uncommon site is capable of ligand binding.

### CysLTs are ago-PAMs for CysLT2R

CysLTs have long been identified as endogenous ligands of CysLTRs[3,4], and it has been widely accepted that they are recognized through the extracellular regions of CysLTRs. In the recently reported LTD4-bound CysLT2R-$G_q$ structure (PDB ID: 9IXX), a cluster of noncontinuous electron density observed at orthosteric binding site was assigned to LTD4[28]. However, at the corresponding position of our structures, CHS molecules were modeled (Figs. 1c, d and 2a). Especially in our LTC4-bound CysLT2R-$G_q$ structure, the unambiguous strong density of CHS in orthosteric binding site is precluded from CysLTs binding. We noticed that LTD4 was only added into the protein complex after size exclusion chromatography (SEC) in that study[28], which maybe account for the different structural characterization. Usually, the GPCR-G protein signaling complex is prepared in the presence of agonist during the whole purification procedure. Furthermore, our CysLTs-bound CysLT2R-$G_q$ structures suggest that the CysLT2R-$G_q$ signaling pathway is synergistically regulated by ceramides and CysLTs through both orthosteric and allosteric mechanisms, which is also implied by their distinct chemical structures (Fig. 1a, Supplementary Fig. 3c). To understand the allosteric regulation mechanism of ceramides by CysLTs, we conducted functional assays to study the cooperativity of these two kinds of ligands. Firstly, we employed NanoBiT-based assays to demonstrate the functional differences between CysLTs and C16:0 ceramide on CysLT2R (Fig. 2b). Consistent with previous findings[18], the potency ($EC_{50}$, half-maximum Effective Concentration) of ceramide (2.749 μM) is significantly lower than that of LTC4 to CysLT2R (3.9 nM). Of interest, LTC4 were observed to give higher maximal G-protein coupling ($E_{max}$, maximum Effect) compared to ceramide, indicating that ceramide functions as a partial agonist for CysLT2R. Next, we assessed LTC4 effects on the signaling behavior of ceramide. The potency ($EC_{50}$) and efficacy ($E_{max}$) of ceramide were significantly enhanced in the presence of LTC4 (Fig. 2c–e, Supplementary Table 3), in line with our prediction, supporting the positive allosteric effects of CysLTs. Meanwhile, we found that a high concentration of ceramide could augment the $E_{max}$ by CysLTs, indicating that the achievement of full activation of CysLT2R needs the presence of both CysLTs and ceramide. Taken together, in addition to acting as agonists on their own, CysLTs also enhance the binding properties of ceramide, thereby being classified as ago-PAMs of CysLT2R.

### Recognition of CysLTs by CysLT2R

The membrane-embedded allosteric binding sites of LTC4 and LTD4 in CysLT2R are largely overlapped (Fig. 3a), which are primarily constituted by a cluster of hydrophobic residues from TM3, TM4, TM5 and ICL2, including Y127[3.41], T130[3.44], V131[3.45], V134[3.48], F137[3.51], L138[3.52],

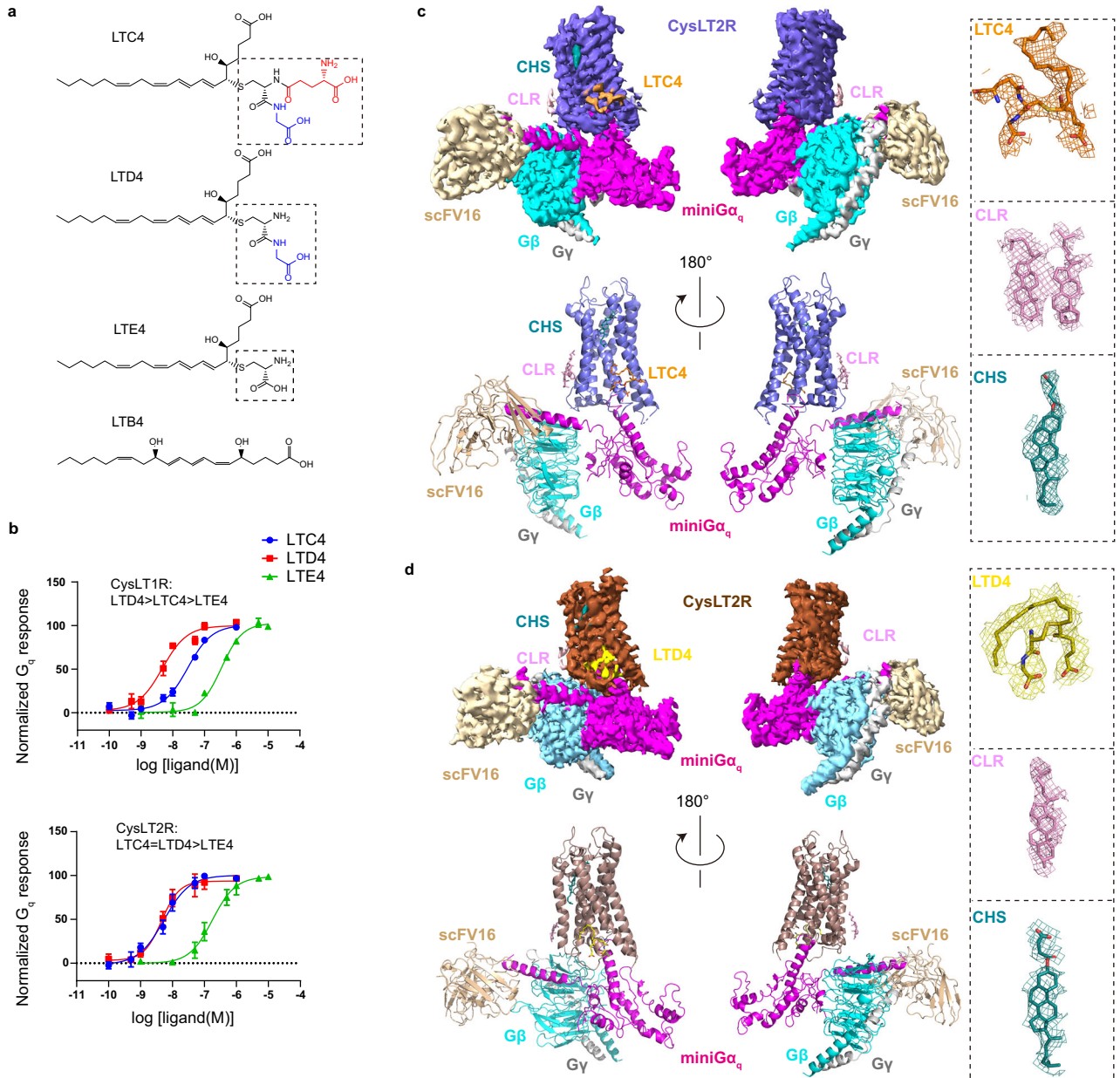

**Fig. 1 | Overall structures of CysLT2R signaling complexes. a** Chemical structures of leukotrienes. Cysteinyl leukotrienes (LTC4, LTD4 and LTE4) contain a peptide portion highlighted in dashed-line rectangle. The amino acids are represented in black (cysteine), blue (glycine) and red (glutamic acid) color. The dihydroxy acid leukotriene LTB4 contains two distinguished hydroxyl moieties. **b** Dose-dependent response curves of CysLT1R and CysLT2R activated by LTC4, LTD4 and LTE4 using NanoBiT-based assays. Data points are presented as mean ± s.e.m. from three independent experiments ($n = 3$). Source data are provided in the Source Data file. **c, d** Cryo-EM density maps (top) and corresponding models (bottom) of LTC4, LTD4-bound CysLT2R-$G_q$. Cryo-EM density maps of protein complex have been shown at a contour level of 0.1–0.18. The density maps for respective ligands and bound molecules in each complex are shown. CHS cholesterol hemisuccinate, CLR cholesterol.

L147[ICL2], L158[4.45], L212[5.49], T216[5.53], C220[5.57], and L223[5.60] (Fig. 3b, c). Most importantly, the negative charged carboxyl head group of glycine of LTC4 and cysteinyl carboxyl group of LTD4 forms a polar interaction network with H142[3.56] stabilized by R145[ICL2], which is crucial for CysLTs induced CysLT2R activation (Fig. 3b, c, Supplementary Fig. 3a). We used a NanoBiT-based assay to examine the contribution of those residues within the pocket. The mutagenesis data showed that mutations of these residues had various effects on receptor activity (Fig. 3d, e). Most mutants exhibited near wild-type expression level (Supplementary Fig. 4g). Interestingly, mutation at residues Y127[3.41], L158[4.45] and L212[5.49], which are distant form LTD4, still affected receptor

activation, indicating the importance of intact structural shape of this binding pocket for receptor activity.

Compared to LTC4, LTD4 occupied a smaller binding pocket with a different orientation of cysteinyl carboxyl group (Fig. 3c). Nevertheless, the polar interaction network with H142[3.56] and R145[ICL2] maintains similar potency for LTD4 as LTC4, highlighting the important role of ICL2 in ligand recognition and GPCR signaling. The structures may explain the lower potency of LTE4, which is attributed to its smaller size and the lack of shape complementarity with the binding pocket. The structure of LTE4-CysLT2R complex is needed to elucidate the precise selectivity mechanism.

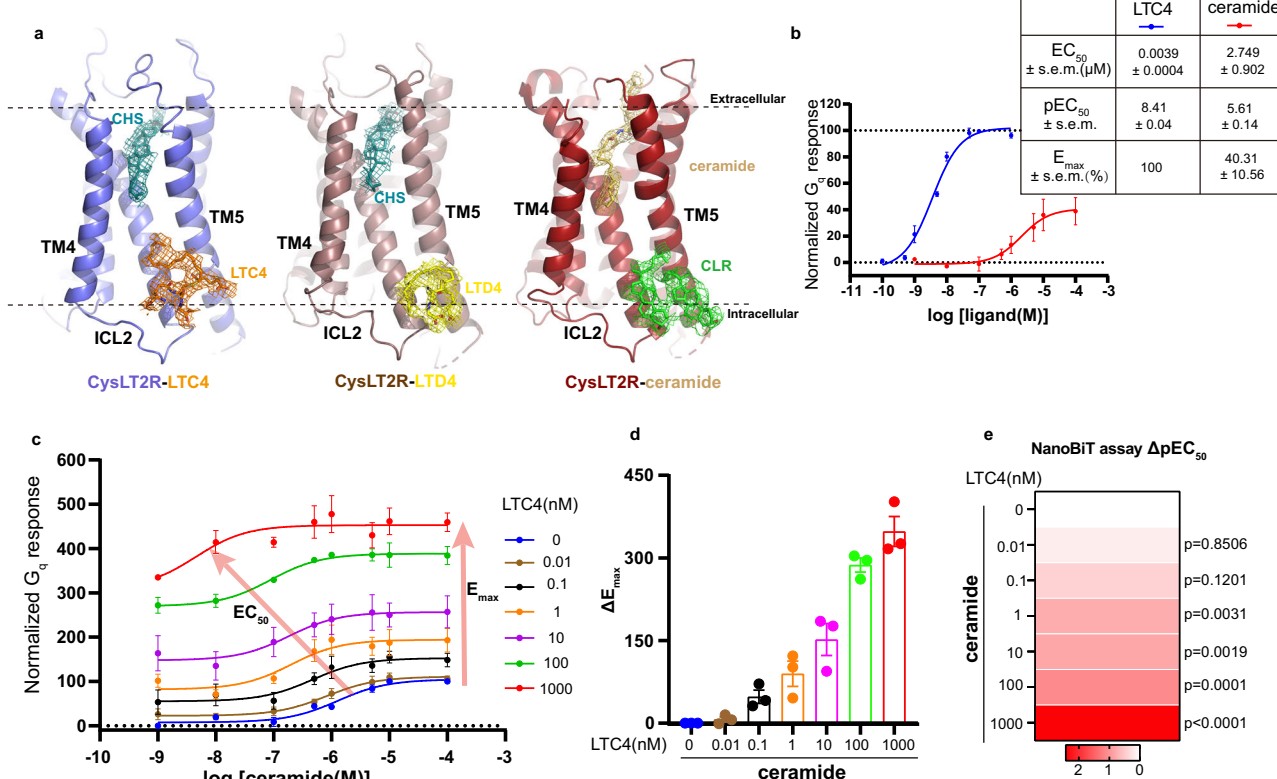

**Fig. 2 | The ago-allosteric modulation at CysLT2R by CysLTs. a** The general positions of LTC4, LTD4, ceramide (PDB ID: 9JH5), CHS and CLR in CysLT2R. Cryo-EM density maps of protein complex have been shown at a contour level of 0.1–0.18. **b** Dose-dependent response curves of CysLT2R activated by LTC4 and C16:0 ceramide. Data points are presented as mean ± s.e.m. from three independent experiments ($n = 3$). Source data are provided in the Source Data file. **c** Concentration-response curves of CysLT2R to the agonist C16:0 ceramide in the presence of different concentrations of LTC4. Data points are presented as mean ± s.e.m. from three independent experiments ($n = 3$). Source data are provided in the Source Data file. **d** The column graphs show $\Delta E_{max}$ ($\Delta E_{max} = E_{max}$ of C16:0 ceramide with different concentrations of LTC4 − $E_{max}$ of C16:0 ceramide

without LTC4), where baseline $E_{max}$ is 100% at 0.1 mM ceramide alone. Data are presented as mean ± s.e.m. from three independent experiments ($n = 3$). Source data are provided in the Source Data file. **e** Concentration-response heatmap of CysLT2R to the agonist C16:0 ceramide in the presence of different concentrations of LTC4. The heatmap is colored according to the $\Delta pEC_{50}$ ($\Delta pEC_{50} = pEC_{50}$ of C16:0 ceramide with different concentrations of LTC4 − $pEC_{50}$ of C16:0 ceramide without LTC4). Each value is the mean ± s.e.m. of three independent experiments ($n = 3$). Statistical significance ($p$ value) was assessed by one-way ANOVA followed by Dunnett's multiple comparisons test, compared with the response of the C16:0 ceramide without LTC4. Source data are provided in the Source Data file.

Interestingly, CHS molecule introduced in purification buffer was observed in the cleft between TM4 and TM5 in two CysLT2R structures, which was occupied by ceramide and antagonists (Figs. 1c, d and 2a, Supplementary Fig. 3b). With the largely studied ligand binding properties at this site, we conducted a study to characterize the effects of CHS on CysLT2R signaling. However, CHS did not play a role in activity modulation in our functional assays. CHS cannot act as an agonist to activate CysLT2R, or inhibit receptor activity as an antagonist like pranlukast (Supplementary Fig. 3d, e). We propose that CHS acted as a placeholder that helped maintain the architecture of extracellular and transmembrane domains to enhance the stability and conformational homogeneity of CysLT2R–$G_q$ complexes, enabling high-resolution structure determination.

### Conformational changes of CysLT2R from inactive to active state

The comparison of our structures with previous reported inactive (11a-bound), intermediate active (11c-bound) and ceramide-activated structures has enabled us to investigate the ago-allosteric activation process of CysLT2R by CysLTs[18,27]. The LTC4-bound and LTD4-bound CysLT2R-$G_q$ complex structures show high overall similarity (Fig. 4d). With a higher resolution, we use the LTC4-bound CysLT2R-$G_q$ structure for further structural comparison.

In the LTC4-bound active structure, the absence of ligand binding in the orthosteric binding pocket released the residues involved in ligand binding, allowing them to undergo conformational changes and enabling free conformation transmission. Structure comparison showed that the significant conformational difference between LTC4-bound and ceramide-bound structures occurred at extracellular part (Fig. 4a, b). Especially, the ECLs undergo large conformational replacement between these two structures (Fig. 4a, b, d). Additionally, the extracellular part of TMs also exhibits markedly different conformations. For detailed instance, the polar interactions between Y119[3.33]-Y263[6.51] and Y123[3.37]-H264[6.52], which are broken upon activation in ceramide-bound structure to induce F260[6.48] into active state, are retained in LTC4-bound structure (Fig. 4c). Surprisingly, structural superposition shows that these residues critical for ceramide-induced activation adopt an inactive conformation in the LTC4-bound state (Fig. 4c). Importantly, the toggle switch F260[6.48] is almost identical in both LTC4-bound and 11a-bound structures (Fig. 4c), indicating that LTC4 employs a noncanonical activation mechanism different from ceramide. Therefore, comparison analysis revealed that the extracellular part of LTC4-bound CysLT2R resembles the 11a-bound inactive state with a higher structural similarity than active state in the ceramide-bound structure exhibited by residue mean square distance (RMSD) value (0.899 Å and 1.099 Å, respectively) (Fig. 4d). The prominent conformational changes upon LTC4 activation occurred at

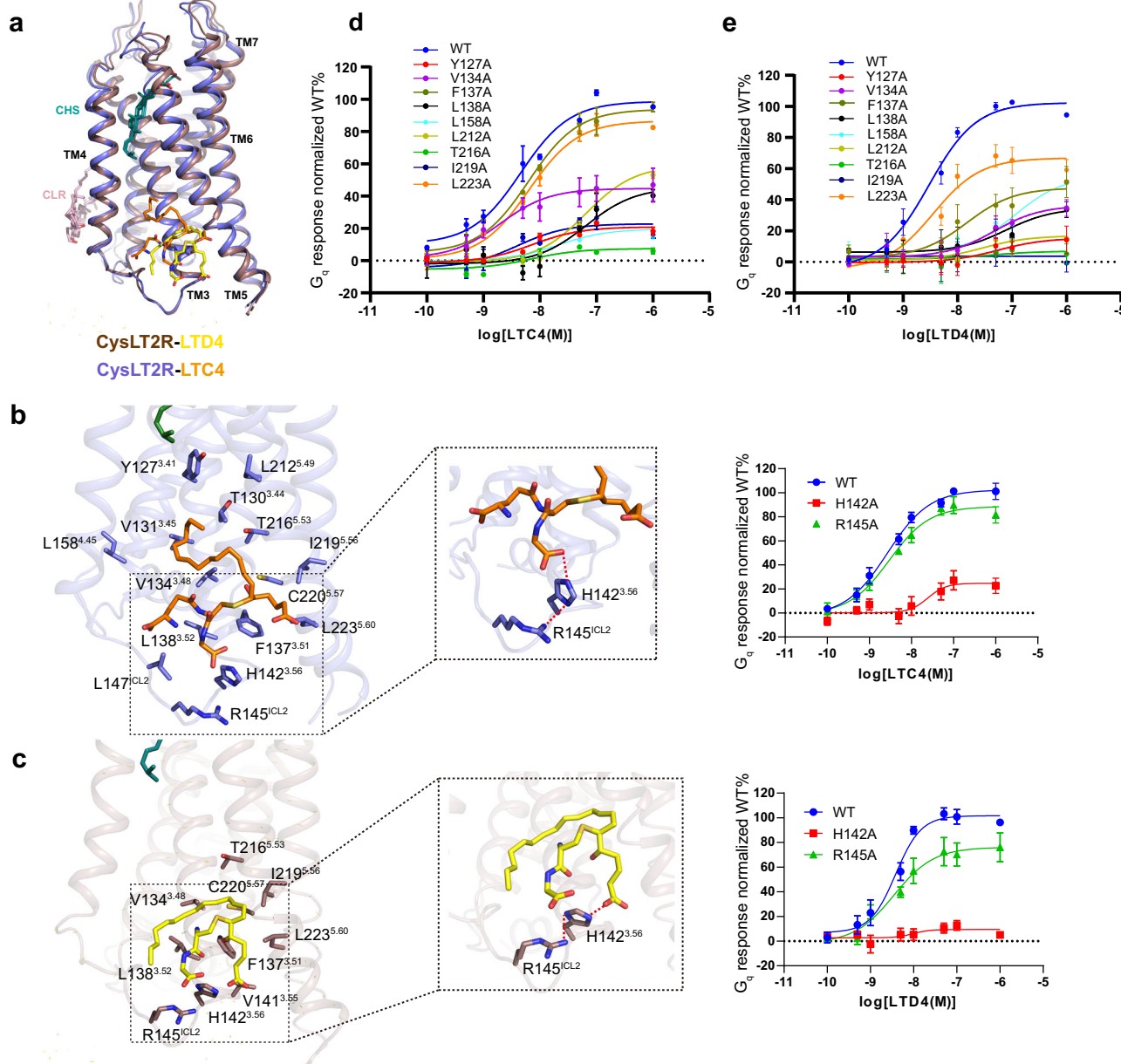

**Fig. 3 | Ligand binding pocket of CysLT2R. a** The alignment of LTC4- and LTD4-bound CysLT2R structures. Details of key residues of the binding pockets are highlighted for LTC4 (**b**) and LTD4 (**c**). Hydrogen-bond interactions are highlighted as red dashed lines. Right panels: concentration–response curves of NanoBiT assay signals of WT and H142[3.56]A and R145[ICL2]A mutants following stimulation with LTC4 (**b**) or LTD4 (**c**). Data are shown as the mean ± s.e.m. from three independent measurements ($n = 3$). Source data are provided in the Source Data file. Effects of mutations of CysLT2R residues in the binding pocket on LTC4 (**d**) or LTD4 (**e**) potency. The pEC$_{50}$ and E$_{max}$ values are provided in Supplementary Table 2. Data are shown as the mean ± s.e.m. from three independent measurements ($n = 3$). Source data are provided in the Source Data file.

cytoplasmic side, suggesting that the activation mechanism CysLT2R by LTC4 begins at the allosteric binding and extends to G$_q$ interface region to facilitate the G$_q$ protein coupling (Fig. 4d, Supplementary Fig. 4a, b). Notably, the intermediate state 11c-bound CysLT2R structure perfectly completes the conformation change trajectory form inactive to active state (Supplementary Fig. 5).

Of note, although the residues Y119[3.33], Y123[3.37], Y263[6.51] and H264[6.52] in orthosteric pocket are not involved in activation propagation path, mutations in these residues significantly affected the receptor activity simulated by LTC4 and LTD4 (Fig. 4c, Supplementary Fig. 4e), indicating the importance of intact structural shape of orthosteric binding pocket for receptor activity.

## Activation mechanisms of CysLT2R by CysLTs

Comparison analysis indicated that LTC4 and LTD4 utilize a distinct propagation path from that found in ceramide-bound structure. A notable event in the allosteric propagation path is the reorganization of ICL2 structure, which is partially disordered in inactive structures (Fig. 5a). The glycine head group of LTC4 forms a polar interaction with H142[3.56], lifting ICL2 to remove clashes between the cytoplasmic ends of TM3 and TM4, ICL2 and Gα$_q$ (Fig. 5a), thus enabling subsequent coupling, and leading to F144[ICL2] projecting into the hydrophobic pocket on Gα$_q$ (Fig. 5a), a feature also found in BLT1 activation[39]. Alanine substitution of H142[3.56], which disrupts the interaction between LTC4 and ICL2, decreased the potency of LTC4 in G$_q$ activation

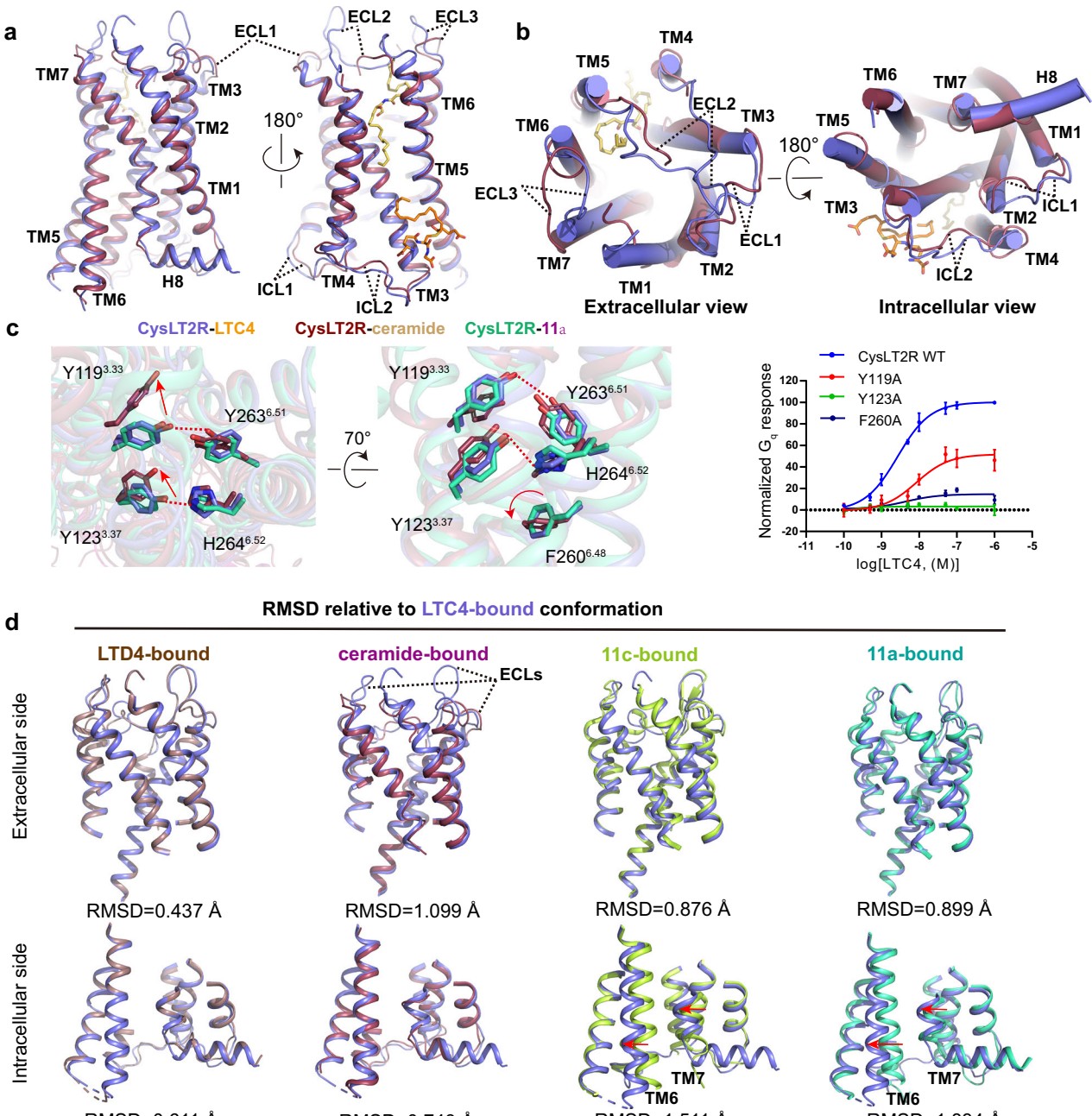

**Fig. 4 | Comparison of CysLT2R structures in different states. a, b** Structure alignment of LTC4 and C16:0 ceramide (PDB ID: 9JH5) activated CysLT2R. **a** Side view; **b** Left: extracellular view; right: intracellular view. **c** Comparison of the structures of CysLT2R bound with LTC4, C16:0 ceramide (PDB ID: 9JH5) and 11a (PDB ID: 6RZ6) at residues Y119[3.33], Y123[3.37], Y263[6.51], H264[6.52] and the "toggle switch" F260[6.48]. Hydrogen bonds are shown as red dashed lines. Data are shown as the

mean ± s.e.m. from three independent measurements ($n = 3$). Source data are provided in the Source Data file. **d** The structural comparison of extracellular and intracellular parts of LTC4- bound with LTD4-, ceramide- (PDB ID: 9JH5), 11c- (PDB ID: 6RZ8) and 11a- (PDB ID: 6RZ6) bound CysLT2R structures. RMSD are presented. The hallmark conformational changes of TM6 and TM7 are marked with red arrows.

(Fig. 3b), highlighting the importance of the upward shift of ICL2 for CysLT2R activation.

The second most significant conformation arrangement occurred at TM5 and TM6 (Fig. 5b). The cysteinyl head group of LTC4 forces TM5 to move towards TM6 by direct contact clash (Fig. 5b, Supplementary Fig. 4c), leading to the reorganization of the interface between TM5 and TM6 to initiate conformational changes that activate the receptor (Fig. 5b). Among the movements of TM5, Y221[5.58] exhibits a striking 'induced-fit' conformational change between the inactive and active states, with its side chain rotating from an outward to a

center position of the TM bundle. This insertion of Y221[5.58] creates a direct steric clash with TM6, leading to the outward shift of TM6. Moreover, Y221[5.58], together with R136[3.50], forms polar interaction network with Y243 of Gα_q to stabilize the active complex (Supplementary Fig. 4d). Consistently, the Y221A mutation abolished LTC4 and LTD4 activity, indicating they employed the same strategy to activate CysLT2R (Fig. 5d, Supplementary Fig. 4e). Interestingly, the Y221F mutant partially recovered the function by substituting a bulky side chain, supporting the crucial role of both Y221[5.58]-R136[3.50]-Gα_q Y243 interactions and the TM5-TM6 relative conformation

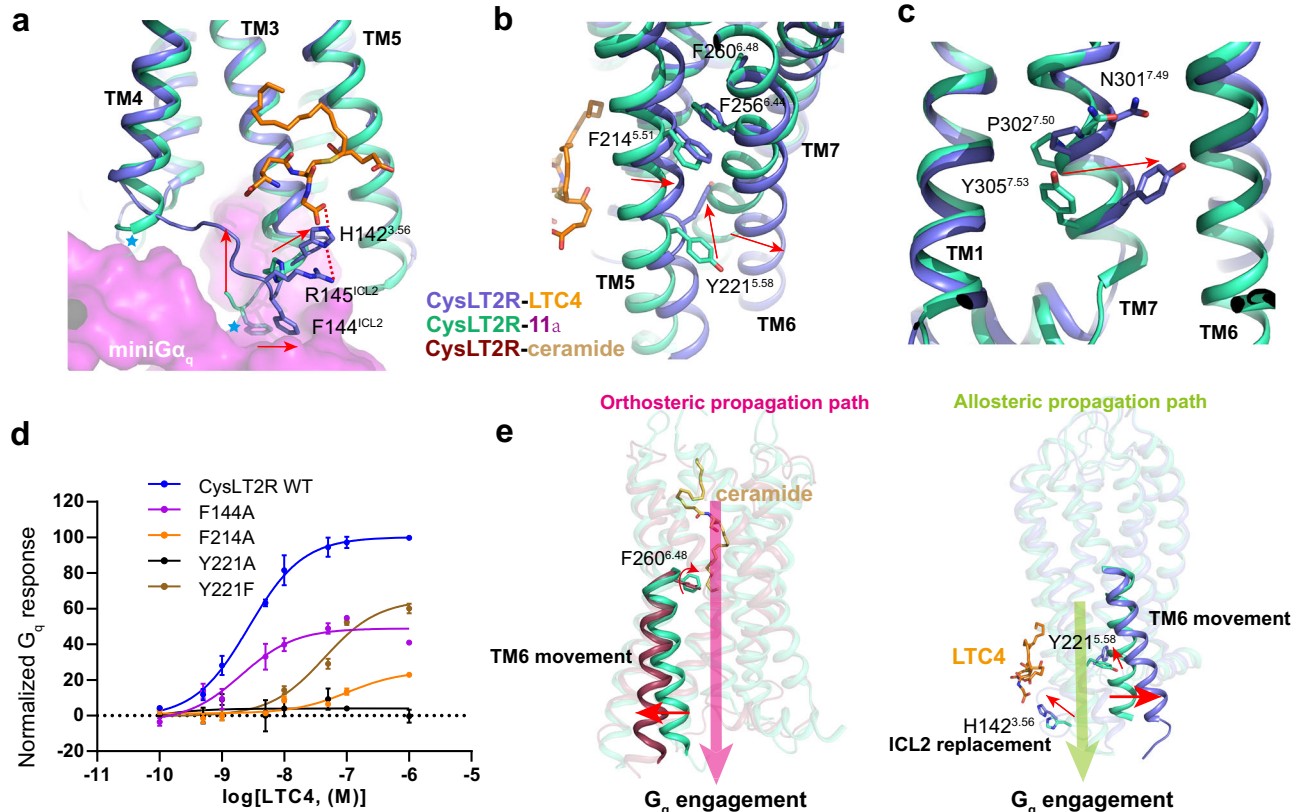

**Fig. 5 | Activation mechanism of CysLT2R.** Conformational differences of important residues and motifs in CysLT2R activation, including residues in ICL2 (**a**), TM5 and TM6 (**b**) and NPxxY motif in TM7 (**c**). The steric clash between inactive CysLT2R and $G_q$ protein is indicated by blue stars. The movement from inactive state to active state is highlighted with red arrows. **d** Effects of mutations in key residues involved in CysLT2R activation by LTC4. The pEC$_{50}$ and E$_{max}$ values are provided in Supplementary Table 2. Data from three independent experiments are presented as the mean ± s.e.m. (*n* = 3). Source data are provided in the Source Data file. **e** Distinct activation propagation paths by ceramide and CysLTs. The orthosteric and allosteric propagation paths are shown by pink and green arrows, respectively. The conformational changes of key residues and the outward movement of TM6 are highlighted with red arrows.

replacement in receptor activation (Fig. 5d, Supplementary Fig. 4e). Additionally, F214$^{5.51}$ also contributes to the reorganization of TM5-TM6 interface (Fig. 5b, d, Supplementary Fig. 4c, e). The residue Y305$^{7.53}$ in NPxxY motif penetrates toward to the center of TM bundle (Fig. 5c), a common feature in class A GPCR activation.

Finally, consistent with the ceramide-bound CysLT2R structure, activation by CysLTs also disrupts a polar interaction network formed by E310$^{8.48}$ and R136$^{3.50}$ and K244$^{6.32}$ (Supplementary Fig. 4f), which blocks the cytoplasmic cavity to maintain the inactive state by restraining the 7TM bundle conformation.

Overall, these results strongly support the noncanonical activation propagation path that the CysLTs-binding at ICL2 site rearranges the intracellular region towards the active conformation without extracellularly induced signal propagation (Fig. 5e). Our structural analysis provided a comprehensive understanding of allosteric agonism in CysLT2R signaling.

## Shared and distinct structural features of CysLT2R in the coupling of $G_q$

Although the extracellular side conformation of LTC4 and ceramide-bound CysLT2R exhibits remarkable differences (RMSD 1.099 Å), the conformation of intracellular side is similar (RMSD 0.743 Å) (Fig. 4d). Thus, structural comparison with ceramide-bound structure showed that there is a small but remarkable shift of the entire $G_q$ heterotrimer relative to the receptor (Fig. 6a). Given the same C-terminal α5 helix of mini-Gα$_q$ used in these two structures, the observed differences in $G_q$-binding modes may imply an intrinsic distinction of orthosteric and

allosteric activation. However, we cannot rule out the possibility that these structural variations were caused by the extra stabilization of Nb35 bound to mini-Gα$_q$ in ceramide-bound structure. Nevertheless, the C-terminal α5 helix of mini-Gα$_q$ inserts into the cytoplasmic cavity of receptor, leading to extensive amphipathic interactions with receptors that constitute the major coupling interface. In our LTC4-CysLT2R-$G_q$ structure, CysLT2R residues R136$^{3.50}$, L238$^{6.26}$ and S241$^{6.29}$ form polar interactions with residues Y243, Q237 and V246 of mini-Gα$_q$ (Fig. 6b), respectively. The absence of direct interaction between H8 and $G_q$ suggests an inconsequential role in CysLT2R-$G_q$ complex formation (Fig. 6b), supported by the previous functional assay with C-terminal truncation at E300[27]. Importantly, mutations of the amino acids involved in polar interactions to alanine, significantly impaired agonist-induced activity (Fig. 6c). Additionally, the residues M140$^{3.54}$, I224$^{5.61}$, L228$^{5.65}$ and A245$^{6.33}$ in CysLT2R form a hydrophobic core with L236, L240 and L245 in mini-Gα$_q$ (Fig. 6d).

As mentioned above, conformation changes in ICL2 upon LTC4 binding play an important role in $G_q$ coupling, and ICL2 contributes another region that interacts with mini-Gα$_q$. Notably, CysLT2R-$G_q$ coupling is stabilized by the insertion of F144$^{ICL2}$ into the hydrophobic cavity formed by residues L34, V79 and F228 of mini-Gα$_q$ (Fig. 6e), which was also found in BLT1-$G_i$ structure[39] and other GPCR-$G_q$ complexes[31,40,41]. The severe clash between ICL2 region and mini-Gα$_q$ is clearly visible if we superimpose the inactive 11a-bound to the LTC4-bound R2 (Fig. 5b). Upon $G_q$ protein coupling, the ICL2 region is conformational replaced and stabilized by the interaction between the backbone carbonyl group of V149$^{ICL2}$ and R31 in mini-Gα$_q$ with an intra-

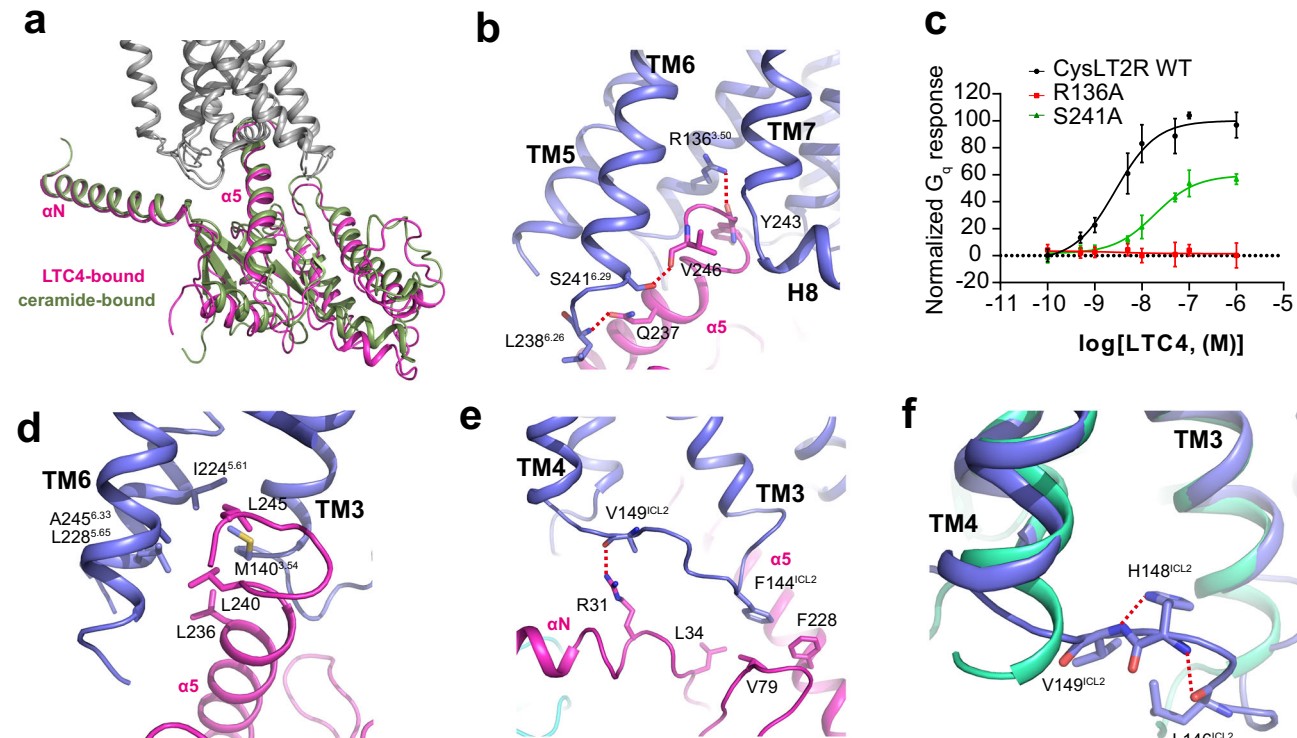

**Fig. 6 | The $G_q$ engagement of CysLT2R. a** Structural superposition of LTC4-bound with ceramide-bound CysLT2R-$G_q$ complex when CysLT2R was aligned. **b, d, e** Detailed interactions between CysLT2R and $G_q$. The α5 of Gα$_q$ is engaged through polar (**b**) and hydrophobic interactions (**d**) by CysLT2R, respectively. Interactions between the ICL2 of CysLT2R (**e**) and $G_q$ shows common and distinct features. Polar interactions are marked with red dashed lines. **c** Representative concentration dependent response curves of mutations in the receptor-$G_q$ interface in response to agonist. Data from three independent experiments are presented as the mean ± s.e.m. ($n = 3$). Source data are provided in the Source Data file. **f** ICL2 structure is stabilized by intra-ICL2 polar interactions upon CysLT2R-$G_q$ coupling. Hydrogen bonds are marked with red dashed lines.

ICL2 polar interaction network formed by H148[ICL2] and backbone of L146[ICL2] and V149[ICL2] (Fig. 6e, f), leading to the unambiguous modeling of L146[ICL2]-H148[ICL2] (Supplementary Fig. 2h, p).

## Disccusion

The cryo-EM structures obtained in this study revealed distinct pockets that accommodates CysLTs with distinct structural features, elucidating the molecular mechanism of ago-PAMs recognition by CysLTRs. Unlike the orthosteric binding mode of LTB4 in BLT1, CysLTs function as ago-PAMs at an allosteric binding site on the membrane surface of TM bundle. In CysLT2R, a direct connection between binding site and $G_q$ coupling has been observed. Instead of inducing the typical sharp kink in TM6 at F[6.48], CysLTs activate CysLT2R without requiring allosteric signal transduction from the extracellular side. Despite extensive efforts to develop CysLT2R-selective antagonists, none have yet reached the market, and their roles in physiological and pathological processes remain less understood. One reason for the unsuccessful drug design may be the lack of agonist-bound CysLT2R structures, which has limited our mechanistic understanding of the ligand recognition and receptor activation. To the best of our knowledge, there has been no discussion about the allosteric binding properties of CysLTs to their receptors. This gap in understanding of CysLTs-CysLTRs signaling axis has misled the physiological and pathological research and obstructed the drug development.

In the past decades, CysLTs and antagonists have been wildly accepted to be recognized by orthosteric site of CysLTRs. Recently, a Cryo-EM structure of LTD4-bound CysLT2R was reported with an orthosteric binding mode, which exhibits significant discrepancies with our structures. For structural study, LTD4 was added after SEC in that paper. In contrast, we solved structures with ligands present throughout purification, ensuring equilibrium binding to the native site. We propose that our experimental approach more faithfully recreates the physiological binding pathway of these lipid mediators. This difference is critical because: Pre-SEC ligands addition. stabilizes the physiological complex, allowing ligands to access membrane-embedded allosteric site during complex formation and detergent solubilization steps. Post-SEC supplementation of ligands apparently cannot access the allosteric site, which remains sequestered within detergent micelles; accordingly, no interpretable density of LTD4 appears at allosteric site (Supplementary Fig. 3f). The weak signal formerly assigned to LTD4 at orthosteric site more plausibly represents CHS introduced in purification buffer (Supplementary Fig. 3g). This observed discrepancies likely stem from methodological variations, underscoring how purification strategies influence the interpretation of GPCR−ligand complexes.

Our findings also provided a potential explanation for the challenges encountered in the development of CysLT2R-targeted therapeutics. The limited efficacy and side effects reported for orthosteric antagonists may stem from their incomplete inhibition of receptor activation. This scenario is especially plausible in light of our finding that potent endogenous agonists−CysLTs and ceramides−activate the receptor via distinct but synergistic loci: the allosteric "ICL2 site" and the orthosteric pocket, respectively. Orthosteric antagonists likely failed to fully quench this coordinated signaling. Furthermore, off-target effects could arise from the high structural conservation of the orthosteric pocket among related GPCRs, leading to suboptimal subtype selectivity. The regulatory effects of diverse molecules bound to "ICL2 site" have been observed in many GPCRs, such as avacopan in C5aR1[42], AP8 and DHA in GPR40[40,43], AZ-1729 and Compound 187 in FFA2[44], LY3154207 in D1R[45], INT-777 in GPBAR[46], compound 6 in β2AR[47]

and cholesterol in GHSR[48]. Some are lipids or sterols, which can align with lipids in bilayers, enhancing the compactness of receptors with membrane environment and stability at the membrane-water interface. Notably, a number of drug-like ligands at "ICL2 site" could control signaling by stabilizing particular transmembrane domain or ICL2 conformations and consequently affecting differential G-protein-subtype selectivity or arrestin coupling[44]. Structural comparison shows that the multiple binding poses of ligands at "ICL2 site" can be further classified into three classes: class 1 without direct interaction with ICL2, class 2 at cavity mainly formed by TM3, TM4 and ICL2, and class 3 at a shallow pocket located between TM3, TM5 and ICL2 (Supplementary Fig. 6a). Ligands in both class 2 and class 3 sites have direct interactions with ICL2. In LTC4-bound CysLT2R, besides the class 3 site occupied by LTC4, we also found a clear class 2 pocket (Supplementary Fig. 6b), indicating a potential strategy of development selective negative allosteric modulators (NAMs) that disrupt the $H142^{3.56}$-$R145^{ICL2}$ interaction network, could achieve more precise and potent inhibition of CysLT2R signaling without orthosteric competition. Alternatively, given the spatial proximity of orthosteric and allosteric sites, designing bitopic ligands that simultaneously engage both sites could harness the cooperative activation mechanism for enhanced efficacy and selectivity. With the emergence of structure-based drug design[49], this structure-based approach paves the way for developing next-generation therapeutics with improved safety and efficacy profiles for diseases driven by CysLT2R hyperactivation.

Looking forward, our structural and mechanistic insights open several transformative paths for therapeutics and translational research. Firstly, we provide a rationale for revisiting clinical trials for CysLT2R-related diseases (e.g., severe asthma, cardiovascular disorders, and specific cancers). Patient stratification based on biomarkers reflecting both CysLTs and ceramides signaling could identify subgroups that would respond optimally to allosteric modulators. Secondly, the ago-PAM function of CysLTs suggests that maximal therapeutic efficacy might be achieved through a dual-targeting strategy using combination therapies or bitopic ligands that block both the orthosteric and allosteric sites simultaneously. Finally, to unequivocally validate the physiological relevance of this allosteric pocket, future work must involve in vivo studies. This includes generating knock-in mouse models with point mutations to disrupt the allosteric site, which would allow us to assess its necessity in disease models. Success in these endeavors would not only confirm the fundamental biology but also firmly establish the allosteric "ICL2 site" as a viable and promising target for next-generation precision medicines.

In summary, our findings clarify the allosteric regulatory roles of CysLTs in CysLTRs signaling pathway, which should be considered in both laboratory research and clinical trials.

## Methods

### Cell lines
*Spodoptera frugiperda* (Sf9, Expression systems) cells were gown in sf-900™ SFM II (ThermoFisher) medium at 27 °C and 500 g. The HEK293T (ATCC, CRL-11268) cells were grown in a humidified 37 °C incubator with 5% CO$_2$ and maintained in Dulbecco's Modified Eagle Medium (DMEM, Gibco) containing 10% fetal bovine serum (FBS, Viva cell) and 1% penicillin-streptomycin.

### Constructs
DNA sequences encoding human CysLT2R was codon-optimized and synthesized for Sf9 insect cell expression and cloned into a pFastBac1 (Invitrogen) vector with an N-terminal FLAG epitope (DYKDDDD) followed by a fragment of β2AR N-terminal tail region (BN)[48] as fusion protein, along with a C-terminal 8× His tag to facilitate the protein expression and purification. The CysLT2R sequence had no additional mutations or loop deletions. A TEV cleavage site was inserted between BN and CysLT2R gene sequences. The prolactin sequence was placed into the N terminus before the FLAG tag as signaling peptide to increase CysLT2R cell membrane localization and increase CysLT2R expression.

A NanoBiT tethering strategy was used to improve sample quality as described previously[35]. The LgBiT subunit was directly fused at the C-terminus of the receptors. miniGα$_q$ were cloned into the pFastBac1 vector. Human Gβ$_1$ with an N-terminal 6× His tag and human Gγ$_2$ were cloned into the pFastBac-Dual vector (Invitrogen). The HiBiT subunit was fused to Gβ$_1$ at the C-terminus with a 15-amino acid linker (GGSGGGGSGGSSSGG).

For functional study, CysLT2R were cloned into the pcDNA3.1 vector (Invitrogen), with a Flag tag added to the receptor's N-terminus. Receptor mutants were generated by site-directed mutagenesis using the Mut Express II Fast Mutagenesis Kit V2 (Vazyme), and the sequences of all constructs were verified (Supplementary Table 4).

### Preparation of scFv16
Coding sequence of scFv16 was constructed into pFastBac1 vector with a GP67 signaling peptide inserted into the N-terminal and a TEV cleavage, 8× His tag at the C-terminal. The purification of scFv16 was conducted as previously described[48]. In brief, secreted scFv16 from Sf9 insect cell culture infected by baculovirus was purified using Ni-NTA and size exclusion chromatography. After balancing the pH and removing the chelating agents by Ni$^{2+}$ and Ca$^{2+}$, the cell culture supernatant was loaded into Ni-NTA (GenScript). The Ni-NTA resin was firstly washed with 20 mM HEPES pH 7.5, 150 mM NaCl, 50 mM imidazole for 10 column volumes and then eluted in buffer containing 250 mM imidazole. The eluted sample was collected and concentrated to 1 ml with a 30 kD MWCO Millipore concentrator and then applied to Superdex™ 200 Increase 10/300 GL column (Cytiva) with buffer containing 20 mM HEPES pH 7.5 and 150 mM NaCl. The monomeric peak fractions were concentrated and fast-frozen by liquid nitrogen.

### Protein complex expression and purification
CysLT2R, miniGα$_q$, Gβ$_1$γ$_2$ and scFv16 were co-expressed in Sf9 insect cells using the Bac-to-Bac baculovirus expression system (Thermo-Fisher) as described previously[48]. The Sf9 cells grown at a density of $4 \times 10^6$ cells per ml were co-infected with separate recombinant baculoviruses at the ratio of 3:3:1:1 (receptor: miniGα$_q$: Gβ$_1$γ$_2$: scFv16.). Forty-eight hours after infection, the cells were collected by centrifugation and stored at −80 °C for further use.

For the purification of protein complex, we performed an anti-FLAG purification followed by size exclusion chromatography to avoided overnight incubation and minimized the time the protein complex spent in solution, thereby significantly reducing dissociation and aggregation, and completed cryo-EM grid preparation within a single day. In brief, cell pellets from 2 L culture were thawed at room temperature and resuspended in lysis buffer (20 mM Tris pH 8.0, 50 mM NaCl, 2 mM CaCl$_2$, 2 mM MgCl$_2$, 0.2 μg/ml leupeptin and 0.2 mg/ml benzamidine). The mixture was stirred at room temperature for 25 min, followed by centrifugation at $60,000 \times g$ for 30 min to collect the precipitate. Then, the mixture was resuspended in formation buffer (20 mM HEPES pH 7.5, 150 mM NaCl, 1 mM CaCl$_2$, 1 mM MgCl$_2$). The complex was formed on membrane in the presence of 50 nM LTC4 or LTD4 (TargetMol) and treated with 25 mU/mL apyrase (Sigma), 2 mg scFv16, followed by incubation for 1.5 h at room temperature. The complex was then solubilized in buffer containing 20 mM HEPES, pH 7.5, 150 mM NaCl, 20% glycerol, 0.75% (w/v) lauryl maltose neopentylglycol (LMNG, Anatrace), 0.1% (w/v) cholesteryl hemisuccinate TRIS salt (CHS, Anatrace), 50 nM LTC4 or LTD4, 25 mU/mL apyrase and 2 mg scFv16 for 1.5 h at 4 °C. The supernatant was collected by centrifugation at $60,000 \times g$ for 30 min, and the solubilized complex was incubated with G1 anti-FLAG resin (GenScript) for 1 h at 4 °C. The resin was collected and washed with 20 column volumes of 20 mM HEPES pH 7.5, 150 mM NaCl, 0.05% (w/v) LMNG, 0.01% (w/v) CHS, 50 nM LTC4

or LTD4, 0.2 μg/ml leupeptin and 0.2 mg/ml benzamidine. The complex was then eluted with 20 mM HEPES pH 7.5, 150 mM NaCl, 0.05% (w/v) LMNG, 0.01% (w/v) CHS, 50 nM LTC4 or LTD4, 0.2 μg/ml leupeptin, 0.2 mg/ml benzamidine, and 0.2 mg/ml FLAG peptide. The complex was collected and concentrated to 0.5 ml with a 100 kD MWCO Millipore concentrator, then loaded onto a Superose 6 Increase 10/300 GL column (Cytiva) with buffer containing 20 mM HEPES, pH 7.5, 150 mM NaCl, 0.001% (w/v) LMNG, 0.0001% (w/v) CHS, 0.00025% (w/v) GDN and 50 nM LTC4 or LTD4. Fractions of the monomeric peak were analyzed by SDS-PAGE and Coomassie staining. Then the desired fractions of CysLT2R-miniG$\alpha_q$ complex were pooled and concentrated for cryo-EM sstudies. The sample was concentrated to ~10 mg/ml for making cryo grids.

## EM sample preparation and data acquisition

For cryo-EM grid preparations, 4 μL of the purified sample was applied to glow-discharged (60 s, SuPro Instruments) Quantifoil R 1.2/1.3 grids (Au 300; Electron Microscopy Sciences). The grids were then plunge-frozen in liquid nitrogen-cooled liquid ethane using a Vitrobot Mark IV (Thermo Fisher Scientific). The Vitrobot was operated at 6 °C, with a blotting time of 4–6 s, using a blot force of '1', and maintaining 100% humidity. Micrographs were collected with Titan Krios microscopes (Thermo Fisher Scientific) operating at 300 kV, equipped with a K3 Summit detector (Gatan) in counting mode (pixel 0.855 Å). Details of all datasets can be found in Supplementary Table S1. All the datasets underwent processing using the similar general workflows, as outlined below.

## Cryo-EM structure determination

Supplementary Fig. 2 shows cryo-EM workflows. Image processing was performed in CryoSPARC v.4[50] and RELION 4.1[51]. Movie stacks were gain-corrected, motion-corrected, and dose-weighted in MotionCor2.1[52]. Contrast transfer function (CTF) estimates were performed using Patch CTF estimation in CryoSPARC v.4, and micrographs with CTF fit resolutions better than 4.0 Å were selected using Curate Exposures in CryoSPARC v.4. Particles were auto-picked using Blob picker in CryoSPARC v.4[50].

For the dataset of CysLT2R-LTC4-$G_q$ complex, 4,024,246 particles were extracted from 5,327 micrographs in CryoSPARC v.4 using a box size of 256 with a binning factor of 3 for ab initio reconstruction and heterogeneous refinements. The particles were subjected to three round s of heterogeneous refinement in CryoSPARC v.4 to remove erroneously picked particles and those particles that did not yield high-resolution reconstructions. After the heterogeneous refinement procedure, 324,384 particles from the class exhibiting clear structural features were selected, reextracted and subjected to non-uniform refinement with C1 symmetry, which yielded a reconstruction at 3.6 Å overall resolution. After one round of Bayesian polishing in RELION 4.1, the particles were imported into CryoSPARC v.4 for heterogeneous refinement which focus on transmembrane domain (TMD). 111,836 particles from the class with the best TMD densities were selected and subjected to CTF (both global and local) and non-uniform refinements in CryoSPARC v.4, resulting in a map at 3.3 Å overall resolution.

For the dataset of CysLT2R-LTD4-$G_q$ complex, 5,785,807 particles were extracted from 6126 micrographs in CryoSPARC v.4 using a box size of 256 with a binning factor of 3 for ab initio reconstruction and heterogeneous refinements. The particles were subjected to three rounds of heterogeneous refinement in CryoSPARC v.4 to remove erroneously picked particles and those particles that did not yield high-resolution reconstructions. After the heterogeneous refinement procedure, 344,699 particles from the class exhibiting clear structural features were selected, reextracted and subjected to non-uniform refinement with C1 symmetry, which yielded a reconstruction at 3.8 Å overall resolution. After one round of Bayesian polishing in RELION 4.1,

the particles were imported into CryoSPARC v.4 for heterogeneous refinement which focus on transmembrane domain (TMD). 130,645 particles from the class with the best TMD densities were selected and subjected to CTF (both global and local) and non-uniform refinements in CryoSPARC v.4, resulting in a map at 3.5 Å overall resolution.

The crystal structures of CysLT2R (PDB code: 6RZ6)[27] were used as initial model for model rebuilding and refinement against EM density map. $G_q$ trimer and scFv16 model was taken from the 5-HT$_{2A}$R–$G_q$ complex (PDB ID: 6WHA)[31]. The model was docked into the EM density map using UCSF Chimera[53], then refined in real space in COOT[54], and further refined in real space using PHENIX[55]. The final models have good stereochemistry and Fourier shell correlations (FSC) with the cryo-EM maps (Supplementary Fig. S2; Supplementary Table S1). Structural Figures were prepared with Pymol (pymol.org), ChimeraX[56], Chimera[53].

## NanoBiT-based G$\alpha_q$-recruitment assay

The recruitment of G$\alpha_q$ to CysLT2R was measured using the Promega NanoBiT Protein-Protein Interaction System. In brief, HEK293T cells seeded at 1.3 ×10[6] cells per well on 6-well plates were co-transfected with pcDNA3.1 plasmids encoding CysLT2R C terminus fusion of SmBiT (2.5 μg) and G$\alpha_q$ construct with N-terminal fusion of LgBiT (2.5 μg) using transfection reagent Lipofectamine 3000 (Thermo-Fisher). After 24 h, cells were harvested using trypsin and then seeded in the flat-bottom white 96-well plate at a density of 100,000 cells per well. This was followed by an overnight incubation, after which experimental assays were conducted.

The growth medium was removed, and NanoBiT assay buffer (1× Hank's Balanced Salt Solution containing 20 mM HEPES at pH 7.5, and 10 μM Furimazine Promega) was added at a 90 μL per well volume. The plate underwent a 20 min dark incubation at 37 °C, followed by a 5 min equilibration at room temperature before starting the readings. Bioluminescence was measured with Synergy H1 multimode reader (BioTek), baseline luminescence was measured 6 times at room temperature until signal stabilization. Subsequently, 10 μL of 10× ligand solution was added, followed by 6 s of vortex mixing. Chemiluminescence was then continuously monitored for 40 min using a multimode plate reader (gain = 140). The data was subsequently analyzed using GraphPad Prism 10.1.1 with a nonlinear regression method.

## Surface expression analysis

Cell surface expression for WT-CysLT2R and various mutants was monitored by a fluorescence-activated cell sorting (FACS) assay (Supplementary Table 4). In brief, HEK293T cells expressing Flag-tagged CysLT2R were harvested 24 h after transfection. Cells were suspended by PBS buffer and blocked with 5% (w/v) BSA at room-temperature for 30 min, then followed by the incubation with mouse anti-Flag-FITC antibody (Proteintech) at a dilution of 1:200 for 30 min at 4 °C, and then PBS buffer was added to cells. Finally, the surface expression of CysLT2R was monitored by detecting the fluorescent intensity of FITC with LSRFortessa (BD). The FACS data were analyzed by FlowJo 10.

## Quantification and statistical analysis

For G$\alpha_q$-recruitment assays, data were normalized to the percentage of the wild-type response and analyzed using the "log(agonist) vs. response" model in GraphPad Prism 10.1.1. For cell surface expression studies, data were normalized to the wild-type response, and processed in GraphPad Prism 10.1.1. All data in Figures and tables are presented as mean ± standard error of the mean (s.e.m.), with the number of biological and technical replicates detailed in the corresponding Figure and table legends, where "$n$" indicates the number of biological replicates.

## Reporting summary

Further information on research design is available in the Nature Portfolio Reporting Summary linked to this article.

## Data availability

The cryo-EM density maps for the LTC4–CysLT2R–$G_q$ and LTD4–CysLT2R–$G_q$ complexes have been deposited in the EMDB under the accession codes EMD-63985 and EMD-63986, respectively. The coordinates for the models of LTC4–CysLT2R–$G_q$ and LTD4–CysLT2R–$G_q$ have been deposited in the PDB under the accession numbers 9UAM and 9UAN, respectively. Source data are provided with this paper.

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

## Acknowledgements

We thank staff members from the Center of Cryo-Electron Microscopy at Southern University of Science and Technology. We thank the Scientific Research Center of Guangzhou Medical University for providing the instrument measurement and analysis. We thank Dr. Jinpeng Sun of Qilu hospital and School of Basic Medical Sciences, Shandong University for kindly providing the model of C16:0 ceramide-CysLT2R-$G_q$ (9JH5) to us for structural analysis. This research was supported by fundings from Guangzhou Medical University Startup (02-412-2302-2141XM, 06-445-1087 and 02-410-2414-3 to H.L.), the National Natural Science Foundation of China (G23111018 to H.L.; 32370046 to C.W.), Science and Technology Projects in Guangzhou (2025A04J7168 to H.L.), the Guangdong Basic and Applied Basic Research Foundation (2024A1515030085 to C.W.) and Startup of Shenzhen Institute of Advanced Technology, Chinese Academy of Sciences (to C.W.).

## Author contributions

H.L. initiated the project for CysLT2R structural investigation. H.L. and C.W. supervised the whole project. M.L., X.B., Y.G., X.M., M.X. S.L. and F.Z. constructed the expression plasmids, purified the protein complex for data collection, performed all the functional assays and data analysis, prepared the Figures and tables, and participated in manuscript editing, supervised by H.L. C.W. and H.L. designed the cryo-EM experiments. W.C., M.L., and W.C. optimized the cryo-EM samples preparation. W.C. prepared the cryo-EM grids, collected cryo-EM images, and performed map calculations under the supervision of C.W. C.W., H.L. and T.C. built and refined the structure models. H.L. prepared the draft of the manuscript, with important input from X.B., J.L. and L.Z. H.L. and C.W. wrote the manuscript.

## Competing interests

The authors declare no competing interests.
