## [Transparent Peer Review file · Nature Communications]

Molecular insights into ago-allosteric modulation at Cysteinyl leukotriene receptor 2

Corresponding Author: Professor Heng Liu

Version 0:

Reviewer comments:

Reviewer #1

(Remarks to the Author)

The study by Li et al. revealed cryo-EM structures of the human CysLT2R–Gq complex in complex with endogenous agonists LTC4 and LTD4. The authors identified a novel allosteric binding site for these ligands and characterized them as ago-PAMs that synergized with orthosteric ceramides. This work demonstrated that CysLTs were recognized within a lipid-facing pocket above ICL2 near the cytoplasmic side of the receptor. Furthermore, they proposed a noncanonical activation mechanism between this allosteric pocket and the Gq-binding site. The findings provided significant mechanistic insights into CysLT2R signaling and offered new perspectives for therapeutic targeting. Overall, I think this is a well-organized paper that advances our understanding of cysteinyl leukotriene receptor structure and function. However, I have several concerns that require clarification and revision before I can recommend publication.

1. The manuscript highlights a significant discrepancy between your LTD4-bound structure (allosteric site) and the very recent structure by Jiang et al. (orthosteric site). While you propose the timing of ligand addition as a potential cause, this requires deeper discussion. Please provide a more detailed speculative analysis of the possible technical and biological reasons for this fundamental difference in the Discussion.
2. While you conclusively show CHS is not a direct modulator, its consistent presence in the orthosteric pocket is intriguing. Could it be acting as a structural stabilizer that facilitates the capture of this specific active conformation? Please discuss the potential role of this detergent molecule in stabilizing the receptor structure observed in your complexes, especially considering the weak density for the extracellular regions.
3. The proposed allosteric propagation path from the ICL2 site to the G protein interface is compelling. The Y221A/F mutagenesis is key evidence. Could you provide additional analysis or commentary on the conservation of this allosteric network (e.g., residues like Y221, R136) in other related lipid-activated GPCRs? This would help establish the broader significance of the proposed mechanism.
4. The proposed activation path (Fig. 5) involves ICL2 reorganization and TM5-TM6 shifts, but some mutants (Y119A, Y123A and F260A) shows a significant loss of efficacy for LTC4 (lines 197–199 and Supplementary Fig. 4e), leaving ambiguity about how allosteric binding propagates signal without orthosteric involvement. Please briefly discuss the potential role of these residues in the allosteric activation pathway, why orthosteric pocket residues influence activity if they aren't part of the propagation path.
5. The study focuses exclusively on Gq coupling. Given that CysLT2R is also reported to signal through Gi/o and other pathways, did you investigate whether this novel allosteric activation mechanism influences G protein coupling preference or bias? A comment on the potential implications of your structural findings for understanding biased signaling at CysLT2R would strengthen the biological relevance of the study.

Reviewer #2

(Remarks to the Author)

This manuscript presents a significant contribution to the field of G protein-coupled receptor (GPCR) biology, particularly

focusing on the Cysteinyl leukotriene receptor 2 (CysLT2R). The study employs cryo-electron microscopy (cryo-EM) to elucidate the structural basis of ago-allosteric modulation by endogenous ligands LTC₄ and LTD₄, revealing a novel membrane-embedded binding site distinct from the orthosteric pocket. The findings challenge previous assumptions about CysLTs binding and activation mechanisms, positioning CysLTs as ago-positive allosteric modulators (ago-PAMs) that synergize with ceramides. The work is timely, given the therapeutic relevance of CysLTRs in inflammatory diseases, cardiovascular disorders, and cancers. The methodological rigor, including functional assays and mutagenesis, strengthens the conclusions. However, several areas could be refined to enhance clarity, impact, and accessibility for a broader audience. Below, I outline the strengths, followed by constructive suggestions for improvement.

Strengths of the Manuscript

- 1. Innovative Structural Insights:** The cryo-EM structures of CysLT2R-Gq complexes bound to LTC₄ and LTD₄ (at 3.3 Å and 3.5 Å resolution) are a major highlight. They uncover an unexpected allosteric binding pocket involving TM3, TM4, TM5, and ICL2, which fundamentally shifts our understanding of ligand recognition.
- 2. Comprehensive Functional Validation:** The integration of NanoBiT-based assays with structural data robustly supports the ago-PAM mechanism. The dose-response curves in Fig. 2 and Fig. 3 quantitatively demonstrate how LTC₄ enhances ceramide efficacy and potency, providing strong evidence for synergistic modulation. The mutagenesis data in Fig. 3d,e further validate key residues, reinforcing the structural observations.
- 3. Therapeutic Implications:** The discussion effectively links the findings to drug design, emphasizing how the allosteric site (e.g., "ICL2 site") could enable selective or biased modulators. This is clinically relevant, given the limitations of current CysLTR antagonists.
- 4. Technical Excellence:** The use of NanoBiT tethering and scFv16 stabilization for cryo-EM sample preparation is commendable, as detailed in the Methods. Supplementary Figures (e.g., Supplementary Fig. 2) provide transparent data processing workflows, enhancing reproducibility.

Constructive Suggestions for Improvement

To elevate the manuscript's impact, consider the following revisions:

1. Enhance Structural Interpretation and Clarity:

While the cryo-EM structures are well-resolved, the description of conformational changes could be more intuitive. For example, in the Results section discussing Fig. 4, the comparison with ceramide-bound structures (PDB: 9J5H) is crucial but somewhat dense. Simplify this by adding a schematic or cartoon overlay in Fig. 4 to illustrate the "noncanonical activation mechanism" more vividly. Currently, Fig. 4 shows RMSD values, but an annotated version highlighting key movements (e.g., TM5-TM6 interface reorganization) would aid readability. Please include arrows or color-coding in the figure to denote specific shifts (e.g., TM6 outward movement).

In Fig. 5, the activation propagation paths are described textually, but the figure could benefit from labels or a flow diagram to map the "distinct activation propagation paths by ceramide and CysLTs". Please add panel (f) showing a simplified model of the pathways to complement the structural details.

2. Expand Methodological Details for Reproducibility:

The Methods section is thorough but could provide more context on optimization steps. For instance, mention any challenges faced during complex purification (e.g., in Supplementary Fig. 1) and how they were overcome. Specifically, detail the criteria for selecting the final particle counts in cryo-EM processing (Supplementary Fig. 2), as this affects resolution claims.

Clarify the rationale for using miniGq chimeras in functional assays. While referenced, a brief justification in the main text would preempt questions about potential artifacts in Gq coupling differences (e.g., in Fig. 6a).

3. Strengthen the Discussion and Broader Implications:

The Discussion aptly notes the "ICL2 site" as a drug target but could delve deeper into why previous drug development failed (e.g., side effects of antagonists like montelukast). Connect this to the structural insights more explicitly—perhaps by proposing specific residue targets (e.g., H1423.56 or R145ICL2) for allosteric inhibitors.

Address the discrepancy with the recent LTD₄-bound structure (PDB ID: 9IXX) more directly. In the Results, it's mentioned that LTD₄ was added post-purification in that study, but a comparative analysis in the Discussion (e.g., using Supplementary Fig. 3) would strengthen the argument for the allosteric model.

Explore therapeutic implications beyond structural biology. For example, how might these findings influence clinical trials for CysLT2R-related diseases? A short paragraph on future directions (e.g., in vivo validation) would add translational value.

4. Improve Language and Presentation:

Some sections, like the Introduction, are comprehensive but occasionally verbose (e.g., background on CysLTs could be condensed). Streamline for conciseness to maintain focus on the study's novelty.

Ensure terminology consistency: For instance, use "ago-PAM" uniformly instead of alternating with "ago-allosteric modulator." Also, define abbreviations like EC50 and Emax upon first use in the main text for clarity.

Proofread for minor grammatical errors (e.g., in Results: "therefor hypothesized" should be "therefore hypothesized").

5. Enhance Data Visualization:

In Fig. 2e and Supplementary Table 3, the heatmap and ΔpEC_{50} values effectively show LTC4-ceramide synergy, but adding error bars or confidence intervals would bolster statistical rigor.

For mutagenesis data (Fig. 3d,e and Supplementary Table 2), consider plotting the surface expression levels alongside activity data to distinguish expression defects from functional impacts, as done in FACS assays.

Reviewer #3

(Remarks to the Author)

The manuscript entitled "Molecular insights into ago-allosteric modulation at Cysteinyl leukotriene receptor 2" (NCOMMS-25-59958) by Liu et al is interesting and the results will aid the field in searching for drug candidates targeting CysLT2R. Yet, the following comments should be addressed to make the manuscript suitable for publication in Nature Communications.

1. Fig 1A-B are referred to in the introduction – the description of data, as shown in Fig1A-B, should be moved to the results section.
2. The authors write "To understand the allosteric regulation mechanism of 127 ceramides by CysLTs, we conducted functional assays to study the cooperativity of these two 128 chemically distinct ligands (Fig. 1a, Supplementary Fig. 3c)". It should be added that Fig. 1a, Supplementary Fig. 3c is referring to the chemical structures of some of these compounds and not the functional assays performed.
3. The authors should include the EC50 and not just the pEC50 in Fig. 2B to aid the understanding to the result-text for the readers.
4. Fig 4A, the manuscript text says "the 180 extracellular part of LTC4-bound CysLT2R resembles an inactive state (RMSD 0.899 Å)" but the figure shows RMSD relative to LTC4-bound conformation and shows that 11a-bound has an RMSD=0.899 Å – the authors should clarify how the manuscript text statement corresponds to what the fig. shows.
5. Clarify if ND stands for not detected or not determined in supplementary Table 2, include some representative plots of the flow cytometry data showing the surface expression of CysLT1R mutants as compared to WT.

Minor points:

- Line 155, change "are distant form LTD4" to "are distant from LTD4"
- The text includes a mixture of past and present tense, keep to past tense
- The text includes a mixture of Gαq and Gq, use Gαq throughout the text

Version 1:

Reviewer comments:

Reviewer #1

(Remarks to the Author)

The authors have addressed my concerns in the revised ms.

Reviewer #2

(Remarks to the Author)

I really appreciate the authors have well addressed all my concerns, and I have no more further comments.

Reviewer #3

(Remarks to the Author)

The authors have made appropriate changes and responded satisfactory to my comments, and I now believe that the revised manuscript "Molecular insights into ago-allosteric modulation at Cysteinyl leukotriene receptor 2" (NCOMMS-25-59958) by Liu et al is suitable for publication in the Nature Communications.

Point-by-point response (NCOMMS-25-59958)

We would like to thank all the reviewers for their thorough evaluation of our manuscript and for their constructive suggestions, which helped us to improve the quality of our work. We addressed all reviewers' concerns by substantially revising the manuscript and adding new figures and discussions. Please find a detailed response to each reviewer's comments below, while all the changes in the manuscript are highlighted in yellow.

Reviewer #1 (Remarks to the Author):

The study by Li et al. revealed cryo-EM structures of the human CysLT₂R–Gq complex in complex with endogenous agonists LTC₄ and LTD₄. The authors identified a novel allosteric binding site for these ligands and characterized them as ago-PAMs that synergized with orthosteric ceramides. This work demonstrated that CysLTs were recognized within a lipid-facing pocket above ICL2 near the cytoplasmic side of the receptor. Furthermore, they proposed a noncanonical activation mechanism between this allosteric pocket and the Gq-binding site. The findings provided significant mechanistic insights into CysLT₂R signaling and offered new perspectives for therapeutic targeting. Overall, I think this is a well-organized paper that advances our understanding of cysteinyl leukotriene receptor structure and function. However, I have several concerns that require clarification and revision before I can recommend publication.

Thank you very much for your positive comments on our study.

1. The manuscript highlights a significant discrepancy between your LTD₄-bound structure (allosteric site) and the very recent structure by Jiang et al. (orthosteric site). While you propose the timing of ligand addition as a potential cause, this requires deeper discussion. Please provide a more detailed speculative analysis of the possible technical and biological reasons for this fundamental difference in the Discussion.

We thank the Reviewer for your very helpful consideration and suggestions. Our speculative analysis, based on our experimental approach and the known biochemistry of GPCR–ligand interactions, is summarized below.

Our cryo-EM structures (LTC₄/LTD₄-bound CysLT₂R–G_q) were solved with ligands present throughout purification, ensuring equilibrium binding to the native site. In contrast, Jiang et al. added LTD₄ after size-exclusion chromatography. This difference is critical because: Pre-SEC ligand addition stabilizes the physiological complex, allowing ligands to access membrane-embedded allosteric site during complex formation and detergent solubilization steps. Post-SEC supplementation of LTD₄ apparently cannot access the allosteric site, which remains sequestered within detergent micelles; accordingly, no interpretable density of LTD₄ appears at allosteric site (Supplementary Fig. 3f). Our functional data support allosteric binding as physiologically relevant: LTC₄/LTD₄ enhance ceramide efficacy (Fig. 2 c-e), implying simultaneous occupancy of allosteric (CysLTs) and orthosteric (ceramide) sites.

We propose that our experimental approach more faithfully recreates the physiological binding pathway of these lipid mediators. Converging evidence from structural, functional, and biochemical analyses strongly supports the biological relevance of LTD4 allosteric binding. The observed discrepancies likely stem from methodological variations, underscoring how purification strategies influence the interpretation of GPCR–ligand complexes. In response, we have expanded the Discussion section in our revised manuscript to provide a more thorough treatment of this issue.

2. While you conclusively show CHS is not a direct modulator, its consistent presence in the orthosteric pocket is intriguing. Could it be acting as a structural stabilizer that facilitates the capture of this specific active conformation? Please discuss the potential role of this detergent molecule in stabilizing the receptor structure observed in your complexes, especially considering the weak density for the extracellular regions.

We thank the Reviewer for this insightful comment. The consistent presence of CHS in the orthosteric pocket across both structures is notable, and we agree it likely plays an important structural role in stabilizing the observed receptor conformation. Our interpretation is summarized below:

Although functional assays confirmed that CHS does not directly modulate CysLT₂R activity as an agonist or antagonist, its well-defined density in the orthosteric pocket suggests a structural stabilizer role. We propose that CHS acts as a placeholder that helps maintain the architecture of extracellular and transmembrane domains during detergent solubilization and cryo-EM grid preparation. This is particularly relevant given the weak density in extracellular regions, suggesting inherent flexibility in the absence of a high-affinity orthosteric ligand.

In our structures, CHS engages primarily with hydrophobic residues in the orthosteric pocket, which may pre-organize the binding site and reduce conformational heterogeneity. By occupying this cavity, CHS likely prevents collapse or distortion of extracellular regions, thereby facilitating resolution of key intracellular features, including the allosteric site and G_q-coupling interface. This role aligns with the known ability of CHS to stabilize other membrane proteins during purification and structural studies. Importantly, CHS did not impede binding of CysLTs to the allosteric site or receptor activation, confirming that its effect is structural rather than functional.

Thus, CHS enhances the stability and conformational homogeneity of CysLT₂R–G_q complexes, enabling high-resolution structure determination. This does not diminish the biological relevance of our structures, as the allosteric binding mode of CysLTs and the activation mechanism are robustly supported by functional data.

3. The proposed allosteric propagation path from the ICL2 site to the G protein interface is compelling. The Y221A/F mutagenesis is key evidence. Could you provide additional analysis or commentary on the conservation of this allosteric network (e.g., residues like Y221, R136) in other related lipid-activated GPCRs? This would help establish the broader significance of the proposed mechanism.

We thank the Reviewer for this helpful consideration and suggestion. We agree that examining

the conservation of key residues in the allosteric network (particularly Y221^{5,58} and R136^{3,50}) across related lipid-activated GPCRs would strengthen the broader significance of our proposed mechanism. We have included this analysis in the manuscript, and our detailed response is below:

In our structure, Y221^{5,58} is a critical residue in the allosteric propagation path, forming a polar interaction with R136^{3,50} and sterically pushing TM6 outward to facilitate G_q coupling. This role is likely generalizable to other lipid GPCRs that couple to G_q. We examined its conservation across lipid-activated GPCRs, it is conserved as Y221^{5,58} and R136^{3,50} across many lipid GPCRs, including CysLT1R, BLT1, FFARs, and P2Y receptors.

While the exact ligand-binding site differs (orthosteric and allosteric), the downstream propagation path involving ICL2 restructuring and TM5-TM6 repacking appears to be a common theme. Our structures now extend this mechanism to membrane-embedded allosteric agonists (CysLTs) that bypass the orthosteric site entirely.

4. The proposed activation path (Fig. 5) involves ICL2 reorganization and TM5-TM6 shifts, but some mutants (Y119A, Y123A and F260A) shows a significant loss of efficacy for LTC₄ (lines 197–199 and Supplementary Fig. 4e), leaving ambiguity about how allosteric binding propagates signal without orthosteric involvement. Please briefly discuss the potential role of these residues in the allosteric activation pathway, why orthosteric pocket residues influence activity if they aren't part of the propagation path.

We thank the Reviewer for raising this important point. It is indeed intriguing that mutations in orthosteric pocket residues (Y119A, Y123A, F260A) reduce signaling efficacy, even though they are not directly involved in the allosteric propagation path. Our interpretation is summarized below:

Although CysLTs activate CysLT2R primarily through an allosteric pathway involving ICL2 reorganization and TM5–TM6 rearrangement, the orthosteric pocket remains structurally coupled to the global activation mechanism. We propose that residues such as Y119, Y123, and F260 contribute to the structural integrity and conformational stability of the receptor, rather than directly participating in allosteric propagation.

Y119 and Y123 form a hydrogen-bond network with Y263 and H264 in the inactive state. Although this network remains largely intact in our LTC₄-bound active structure, its disruption via mutagenesis may destabilize the extracellular domain, indirectly altering the energy landscape for activation. This is consistent with the general role of orthosteric residues in maintaining GPCR fold and dynamics. F260, part of the conserved CWxP motif, serves as a hydrophobic toggle switch in many class A GPCRs. Its mutation likely perturbs the equilibrium between inactive and active states—even when activation is triggered allosterically.

Critically, the allosteric pathway does not operate in isolation but is integrated within the receptor's broader conformational landscape. The orthosteric pocket may act as a structural scaffold that fine-tunes the activation energy barrier. This explains why its disruption attenuates—but does not abolish—signaling, as observed in our mutagenesis data.

In summary, although allosteric ligands (LTC₄/LTD₄) bypass the orthosteric site for binding, the structural integrity of the orthosteric pocket remains essential for optimal signal transmission.

5. The study focuses exclusively on G_q coupling. Given that CysLT2R is also reported to signal through G_{i/o} and other pathways, did you investigate whether this novel allosteric activation mechanism influences G protein coupling preference or bias? A comment on the potential implications of your structural findings for understanding biased signaling at CysLT2R would strengthen the biological relevance of the study.

We thank the Reviewer for raising this important point about potential G protein coupling bias. We agree that investigating whether the allosteric activation mechanism influences signaling bias would significantly enhance the biological relevance of our findings.

Our structures reveal that allosteric binding of CysLTs induces a distinct conformational change in ICL2 and TM5-TM6, which directly engages G_{α_q}. Notably, the rearrangement of ICL2 is a feature shared with other G_q-coupled GPCRs. This suggests that the allosteric site may be optimized for G_q coupling.

However, whether this mechanism preferentially promotes G_q over G_{i/o} coupling remains an open question. As delineated in the Discussion section, “In LTC₄-CysLT2R-miniG_q, besides the class 3 site occupied by LTC₄, we also found a clear class 2 pocket (Supplementary Fig. 6b), indicating a potential strategy of development selective or biased PAMs of CysLT2R that target this cavity, where accommodates various G protein selective and biased ligands in FFA2”. The ICL2 site emerges as a critical determinant of G-protein selectivity, with therapeutic implications for designing pathway-specific modulators.

Therefore, to comprehensively investigate signaling bias, our future studies will employ NanoBiT-based assays or BRET biosensors to evaluate the activation of G_q and G_i as well as arrestin recruitment mediated by wild-type and mutant CysLT2R in response to LTC₄ and LTD₄ stimulation. Additionally, structural studies of CysLT2R in complex with G_{i/o} protein will be conducted.

Reviewer #2 (Remarks to the Author):

This manuscript presents a significant contribution to the field of G protein-coupled receptor (GPCR) biology, particularly focusing on the Cysteinyln leukotriene receptor 2 (CysLT2R). The study employs cryo-electron microscopy (cryo-EM) to elucidate the structural basis of ago-allosteric modulation by endogenous ligands LTC₄ and LTD₄, revealing a novel membrane-embedded binding site distinct from the orthosteric pocket. The findings challenge previous assumptions about CysLTs binding and activation mechanisms, positioning CysLTs as ago-positive allosteric modulators (ago-PAMs) that synergize with ceramides. The work is timely, given the therapeutic relevance of CysLTRs in inflammatory diseases, cardiovascular disorders, and cancers. The methodological rigor, including functional assays and mutagenesis, strengthens the conclusions. However, several areas could be refined to enhance clarity, impact, and accessibility for a broader audience. Below, I outline the strengths, followed by constructive suggestions for improvement.

We thank the Reviewer for the positive comments and constructive suggestions on our study.

Strengths of the Manuscript

1. **Innovative Structural Insights:** The cryo-EM structures of CysLT2R-Gq complexes bound to LTC4 and LTD4 (at 3.3 Å and 3.5 Å resolution) are a major highlight. They uncover an unexpected allosteric binding pocket involving TM3, TM4, TM5, and ICL2, which fundamentally shifts our understanding of ligand recognition.
2. **Comprehensive Functional Validation:** The integration of NanoBiT-based assays with structural data robustly supports the ago-PAM mechanism. The dose-response curves in Fig. 2 and Fig. 3 quantitatively demonstrate how LTC4 enhances ceramide efficacy and potency, providing strong evidence for synergistic modulation. The mutagenesis data in Fig. 3d,e further validate key residues, reinforcing the structural observations.
3. **Therapeutic Implications:** The discussion effectively links the findings to drug design, emphasizing how the allosteric site (e.g., "ICL2 site") could enable selective or biased modulators. This is clinically relevant, given the limitations of current CysLTR antagonists.
4. **Technical Excellence:** The use of NanoBiT tethering and scFv16 stabilization for cryo-EM sample preparation is commendable, as detailed in the Methods. Supplementary Figures (e.g., Supplementary Fig. 2) provide transparent data processing workflows, enhancing reproducibility.

We thank the Reviewer for the appreciation on our study.

Constructive Suggestions for Improvement

To elevate the manuscript's impact, consider the following revisions:

1. Enhance Structural Interpretation and Clarity:

While the cryo-EM structures are well-resolved, the description of conformational changes could be more intuitive. For example, in the Results section discussing Fig. 4, the comparison with ceramide-bound structures (PDB: 9J5H) is crucial but somewhat dense. Simplify this by adding a schematic or cartoon overlay in Fig. 4 to illustrate the "noncanonical activation mechanism" more vividly. Currently, Fig. 4 shows RMSD values, but an annotated version highlighting key movements (e.g., TM5-TM6 interface reorganization) would aid readability. Please include arrows or color-coding in the figure to denote specific shifts (e.g., TM6 outward movement).

Following the constructive suggestion of the Reviewer, we have included new subpanels Fig. 4a and b to describe the overall conformational changes between our LTC4-CysLT2R-Gq and ceramide-bound structure (PDB: 9JH5). We have already highlighted the loops with large conformational changes by black dash lines. Additionally, we also used red arrows to show the replacement of TM6 and TM7 during activation in Fig. 4d.

In Fig. 5, the activation propagation paths are described textually, but the figure could benefit from labels or a flow diagram to map the "distinct activation propagation paths by ceramide and CysLTs". Please add panel (f) showing a simplified model of the pathways to complement the structural details.

Following the constructive suggestion of the Reviewer, we have reorganized the Fig. 5 and included a new subpanel Fig. 5e to describe the distinct activation propagation paths by CysLTs and ceramide. In this new Fig. 5e, we represent the crucial structure determinants in both activation paths.

2. Expand Methodological Details for Reproducibility:

The Methods section is thorough but could provide more context on optimization steps. For instance, mention any challenges faced during complex purification (e.g., in Supplementary Fig. 1) and how they were overcome. Specifically, detail the criteria for selecting the final particle counts in cryo-EM processing (Supplementary Fig. 2), as this affects resolution claims.

We thank the Reviewer for the constructive suggestion. Actually, we faced challenges in purifying the complex samples. The complex samples were not stable and tended to aggregate and dissociate when the purification lasted two days. So, we tried to streamline the protocol to finish the protein purification and cryo-EM grid preparation within one day. Owing to the high quality of protein sample, we performed the cryo-EM experiment following standard procedures without encountering any obstacles.

Following the suggestion of this Reviewer, we have added the following to the text in Methods section:

“For the purification of protein complex, we performed an anti-FLAG purification followed by size exclusion chromatography to avoid overnight incubation and minimized the time the protein complex spent in solution, thereby significantly reducing dissociation and aggregation, and completed cryo-EM grid preparation within a single day. In brief, cell pellets from 2 L culture were thawed at room temperature [...].”

We also modified the description of cryo-EM data processing:

“After the heterogeneous refinement procedure, 324,384 particles from the class exhibiting clear structural features were selected, [...]. 111,836 particles from the class with the best TMD densities were selected and subjected to CTF (both global and local) and non-uniform refinements in CryoSPARC v.4, resulting in a map at 3.3 Å overall resolution.”

“After the heterogeneous refinement procedure, 344,699 particles from the class exhibiting clear structural features were selected, [...]. 130,645 particles from the class with the best TMD densities were selected and subjected to CTF (both global and local) and non-uniform refinements in CryoSPARC v.4, resulting in a map at 3.5 Å overall resolution.”

Clarify the rationale for using miniGα_q chimeras in functional assays.

We apologize for not being clear enough in our Methods section. We used wild-type Gα_q in

NanoBiT-based $G\alpha_q$ -recruitment assay and we have modified the description in Methods section:

“In brief, HEK293T cells seeded at 1.3×10^6 cells per well on 6-well plates were co-transfected with pcDNA3.1 plasmids encoding CysLT2R C terminus fusion of SmBiT (2.5 μ g) and wild-type $G\alpha_q$ construct with N-terminal fusion of LgBiT (2.5 μ g) using transfection reagent Lipofectamine 3000 (ThermoFisher).”

While referenced, a brief justification in the main text would preempt questions about potential artifacts in G_q coupling differences (e.g., in Fig. 6a).

Following the suggestion of this Reviewer, we have added the following to the text and cited a paper (ref 34) by Nehme, R. et al., in which they extended the family of mini-G proteins to determine the structure of relevant GPCRs in a fully active state:

“[...] which has been successfully used to obtain cryo-EM structures of several other G_q -coupled GPCRs³¹⁻³³. The engineered mini- $G\alpha_q$ construct exhibits functional activity comparable to that of wild-type $G\alpha_q$ ³¹. Owing to its enhanced tendency for forming stable complexes with GPCRs, the mini- $G\alpha_q$ variant, whose C-terminal $\alpha 5$ helix remains unchanged, effectively mimics wild-type $G\alpha_q$ in receptor coupling. Consequently, mini-G proteins have been serving as ideal tools for biophysical investigations of GPCRs in their active state³⁴. To improve the stability of CysLT2R- G_q complexes for cryo-EM studies, [...]”

In addition, we propose that the structural variations of G_q coupling between our structure and ceramide-bound structure (9JH5) may be due to two main reasons: First, the distinct activation modes of receptor lead to the G_q -coupling differences. Second, compared to utilizing scFV16 to stabilize the complex in our purification strategy, ceramide-bound structure was stabilized by scFV16 and Nb35 concurrently. It’s likely that the direct binding to $G\alpha_q$ of Nb35 induced the different conformation. We have modified the text accordingly:

“Thus, structural comparison with ceramide-bound structure showed that there is a small but remarkable shift of the entire G_q heterotrimer relative to the receptor (Fig. 6a). Given the same C-terminal $\alpha 5$ helix of different chimeric mini- $G\alpha_q$ used in these two structures, the observed differences in G_q -binding modes may imply an intrinsic distinction of orthosteric and allosteric activation. However, we cannot rule out the possibility that these structural variations were caused by the extra stabilization of Nb35 bound to mini- $G\alpha_q$ in ceramide-bound structure.”

3. Strengthen the Discussion and Broader Implications:

The Discussion aptly notes the "ICL2 site" as a drug target but could delve deeper into why previous drug development failed (e.g., side effects of antagonists like montelukast). Connect this to the structural insights more explicitly—perhaps by proposing specific residue targets (e.g., H1423.56 or R145ICL2) for allosteric inhibitors.

We thank the Reviewer for this constructive suggestion. We agree that connecting our structural insights to the historical challenges in drug development significantly strengthens the impact of our work.

We propose that the failure of previous orthosteric-based drug development may be due to

two main reasons: (1), Incomplete inhibition: Orthosteric antagonists cannot fully inhibit receptor activation driven by the synergistic action of ceramides (orthosteric) and CysLTs (allosteric), as revealed by our functional data. (2), Lack of selectivity: The orthosteric pocket is structurally conserved among related receptors, potentially leading to off-target effects.

Our structures now provide a clear solution to these challenges. We explicitly highlight the allosteric pocket residues H142 and R145—validated by our mutagenesis studies as critical for agonist function—as prime targets for a new class of drugs. We suggest that developing negative allosteric modulators (NAMs) aimed at this pocket could achieve more complete and selective inhibition of CysLT2R signaling. This structural insight offers a direct and actionable path for rational drug design, moving beyond the limitations of traditional orthosteric antagonists.

We have addressed this point in the third paragraph of the Discussion section.

Address the discrepancy with the recent LTD4-bound structure (PDB ID: 9IXX) more directly. In the Results, it's mentioned that LTD4 was added post-purification in that study, but a comparative analysis in the Discussion (e.g., using Supplementary Fig. 3) would strengthen the argument for the allosteric model.

We thank the Reviewer for your very helpful consideration and suggestion. We contend that our experimental strategy more faithfully recapitulates the physiological binding pathway of these lipid mediators. The observed discrepancies most likely originate from methodological differences, highlighting the decisive influence of purification schemes on the interpretation of GPCR–ligand complexes.

We have addressed this point with Supplementary Fig. 3f and g in the second paragraph of the Discussion section.

Explore therapeutic implications beyond structural biology. For example, how might these findings influence clinical trials for CysLT2R-related diseases? A short paragraph on future directions (e.g., in vivo validation) would add translational value.

We thank the Reviewer for raising this critical point regarding the translational impact of our work. We have now added a dedicated paragraph to address the therapeutic implications and future directions in the fourth paragraph of Discussion section.

4. Improve Language and Presentation:

Some sections, like the Introduction, are comprehensive but occasionally verbose (e.g., background on CysLTs could be condensed). Streamline for conciseness to maintain focus on the study's novelty.

Following the suggestion of the Reviewer, we have condensed the background on CysLTs and start the Introduction section with the description of receptors.

Ensure terminology consistency: For instance, use "ago-PAM" uniformly instead of alternating

with "ago-allosteric modulator." Also, define abbreviations like EC50 and Emax upon first use in the main text for clarity.

We thank the Reviewer for reminding us of this point, as we have checked the consistency and defined all abbreviations.

Proofread for minor grammatical errors (e.g., in Results: "therefor hypothesized" should be "therefore hypothesized").

We appreciate this point as well and have corrected this error. We have proofread the entire manuscript and corrected all grammatical errors.

5. Enhance Data Visualization:

In Fig. 2e and Supplementary Table 3, the heatmap and ΔpEC_{50} values effectively show LTC₄-ceramide synergy, but adding error bars or confidence intervals would bolster statistical rigor.

Following the suggestion of this Reviewer, we have performed statistical significance testing in Fig. 2e and Supplementary Table 3.

For mutagenesis data (Fig. 3d, e and Supplementary Table 2), consider plotting the surface expression levels alongside activity data to distinguish expression defects from functional impacts, as done in FACS assays.

We thank the Reviewer for the constructive suggestion. We have added the surface expression levels plots in Supplementary Fig. 4g.

Reviewer #3 (Remarks to the Author):

The manuscript entitled "Molecular insights into ago-allosteric modulation at Cysteinyl leukotriene receptor 2" (NCOMMS-25-59958) by Liu et al is interesting and the results will aid the field in searching for drug candidates targeting CysLT₂R. Yet, the following comments should be addressed to make the manuscript suitable for publication in Nature Communications.

We thank this Reviewer for the detailed and positive evaluation of our work.

1, Fig 1A-B are referred to in the introduction – the description of data, as shown in Fig1A-B, should be moved to the results section.

We thank the Reviewer for reminding us of this point, as we have removed Fig. 1a, b and the description to Results section.

2, The authors write "To understand the allosteric regulation mechanism of 127 ceramides by CysLTs, we conducted functional assays to study the cooperativity of these two 128 chemically

distinct ligands (Fig. 1a, Supplementary Fig. 3c)". It should be added that Fig. 1a, Supplementary Fig. 3c is referring to the chemical structures of some of these compounds and not the functional assays performed.

We agree with the Reviewer that the description was not clear, and we have modified our description:

"Furthermore, our CysLTs-bound CysLT2R-G_q structures suggest that the CysLT2R-G_q signaling pathway is synergistically regulated by ceramides and CysLTs through both orthosteric and allosteric mechanisms, which is also implied by their distinct chemical structures (Fig. 1a, Supplementary Fig. 3c). To understand the allosteric regulation mechanism of ceramides by CysLTs, we conducted functional assays to study the cooperativity of these two kinds of ligands. Firstly, [...]"

3, The authors should include the EC₅₀ and not just the pEC₅₀ in Fig. 2B to aid the understanding to the result-text for the readers.

We thank the Reviewer for the suggestion. As requested, we have added EC₅₀ in Fig. 2b.

4, Fig 4A, the manuscript text says "the 180 extracellular part of LTC₄-bound CysLT2R resembles an inactive state (RMSD 0.899 Å)" but the figure shows RMSD relative to LTC₄-bound conformation and shows that 11a-bound has an RMSD=0.899 Å – the authors should clarify how the manuscript text statement corresponds to what the fig. shows.

Following the suggestion of this Reviewer and Reviewer #2, we have reorganized Fig. 4 and modified our description to avoid any confusion:

"Therefore, comparison analysis revealed that the extracellular part of LTC₄-bound CysLT2R resembles the 11a-bound inactive state with a higher structural similarity than active state in the ceramide-bound structure exhibited by residue mean square distance (RMSD) value (0.899 Å and 1.099 Å, respectively) (Fig. 4d). The prominent conformational changes upon LTC₄ activation occurred at cytoplasmic side, [...]"

5. Clarify if ND stands for not detected or not determined in supplementary Table 2, include some representative plots of the flow cytometry data showing the surface expression of CysLT1R mutants as compared to WT.

ND stands for "not detected" in supplementary Table 2, as the activation level is too low to detect.

We thank the Reviewer for the constructive suggestion. We have added the surface expression levels plots in Supplementary Fig. 4g.

Minor points:

- Line 155, change "are distant form LTD4" to "are distant from LTD4"

We have changed it.

- The text includes a mixture of past and present tense, keep to past tense

Thank you for pointing out the inconsistent use of tenses. We have now gone through the entire manuscript and made corrections.

- The text includes a mixture of $G\alpha_q$ and G_q , use $G\alpha_q$ throughout the text

We have checked the use of $G\alpha_q$ and G_q in entire manuscript, and made changes to avoid any confusion.